# A Central Asia Hydrologic Monitoring Dataset for Food and Water Security Applications in Afghanistan

Amy L. McNally[1,2,3], Jossy Jacob[1,4], Kristi Arsenault[1,2], Kimberly Slinski[1,5], Daniel P. Sarmiento[1,2], Andrew Hoell[6], Shahriar Pervez[7], James Rowland[8], Mike Budde[8], Sujay Kumar[1], Christa Peters-Lidard[1], James P. Verdin[3]

NASA Goddard Space Flight Center, Greenbelt, MD, 20771, United States
Science Applications International Corporation Inc., Reston, VA, 20190, United States
U.S. Agency for International Development, Washington, DC, 20523, United States
Science Systems and Applications Inc., Lanham, MD, 20706, United States
University of Maryland Earth Systems Science Interdisciplinary Center, College Park, MD, 20740, United States
National Oceanic and Atmospheric Administration, Physical Science Laboratory, Boulder, CO, 80305, United States
Arctic Slope Regional Corporation Federal Data Solutions, Contractor to U.S. Geological Survey, Earth Resources Observation and Science (EROS) Center, Sioux Falls, SD, 57198, United States
U.S. Geological Survey, EROS Center, Sioux Falls, South Dakota, 57198, United States

*Correspondence to*: Amy L. McNally (amy.l.mcnally@nasa.gov)

**Abstract**
From the Hindu Kush Mountains to the Registan desert, Afghanistan is a diverse landscape where
droughts, floods, conflict, and economic market accessibility pose challenges for agricultural
livelihoods and food security. The ability to remotely monitor environmental conditions is critical to
support decision making for humanitarian assistance. The Famine Early Warning Systems Network
(FEWS NET) Land Data Assimilation System (FLDAS) global and Central Asia data streams
provide information on hydrologic states for routine integrated food security analysis. While
developed for a specific project, these data are publicly available and useful for other applications
that require hydrologic estimates of the water and energy balance. These two data streams are
unique because of their suitability for routine monitoring, as well as a historical record for
computing relative indicators of water availability. The global stream is available at ~1 month
latency, monthly average outputs on a 10-km grid from 1982-present. The second data stream,
Central Asia (30-100 °E, 21-56 °N), at ~1 day latency, provides daily average outputs on a 1-km
grid from 2000-present. This paper describes the configuration of the two FLDAS data streams,
background on the software modeling framework, selected meteorological inputs and parameters,
and results from previous evaluation studies. We also provide additional analysis of precipitation
and snow cover over Afghanistan. We conclude with an example of how these data are used in
integrated food security analysis. For use in new and innovative studies that will improve
understanding of this region, these data are hosted by U.S. Geological Survey data portals and the
National Aeronautics and Space Administration (NASA). The Central Asia data described in this
manuscript can be accessed via the NASA repository at 10.5067/VQ4CD3Y9YC0R, the global data
described in this manuscript can be accessed via the NASA repository at 10.5067/5NHC22T9375G.
**1 Introduction**
From the Hindu Kush Mountains to the Registan desert, Afghanistan is a diverse landscape where
droughts, floods, conflict, and economic market accessibility pose challenges for agricultural
livelihoods and food security. The ability to remotely monitor environmental conditions is critical to
support decision making for economic development, humanitarian assistance, water resource
management, agriculture and more. Environmental datasets can be combined with socio-economic
variables and transformed into customized products to support decision making. This is the
definition of a 'climate service' (Hewitt et al., 2012).

Hydrologic and land surface datasets are particularly relevant for agriculture and water resources
decision making. When these datasets are credible, updated routinely, and made publicly available,
the influences of climate variability and climate change can be incorporated into specialized
analyses by intermediary users[1]. One example of an intermediary user central to this data descriptor
is the food security analysts of the Famine Early Warning Systems Network (FEWS NET). FEWS

---

[1] The WMO defines intermediate (intermediary) users as those who transform climate information into a climate service

NET analysts combine environmental information, largely from remote sensing and earth system models, with information on nutrition, livelihoods, markets, and trade to provide decision support to the U.S. Agency for International Development (USAID) Bureau of Humanitarian Assistance. Additional examples and discussion of the production of climate service inputs can be found in the literature (e.g., Vincent et al., 2018; McNally et al., 2019).

While these data are tailored to specific needs, they are also applicable to other climate services and research e.g., Desert Locusts movement forecasting (Tabar et al., 2021). To that end, this paper describes the FEWS NET Land Data Assimilation System (FLDAS) global and Central Asia data streams. The inputs (e.g., precipitation) and resulting hydrologic estimates (a) provide a 40+ year historical record for contextualizing estimates in terms of departures from average (i.e., anomalies), (b) are low latency (< 1-month) for timely decision support, and (c) are familiar to the food and water security user-community.

The purpose of this data descriptor is four-fold:
- to describe the development of the moderate resolution, low latency FLDAS hydrologic monitoring system for Central Asia, specifically Afghanistan
- to increase awareness of these data resources, which are intended to be a public good,
- to demonstrate how our methods inform critical investigations that ultimately improve general understanding of water resources in this important region of the world, and
- to describe a 'convergence of evidence' approach to hydrologic monitoring in locations where all sources of information contain some level of uncertainty.

An outline of this data descriptor is as follows. Section 1.1 provides background on Afghanistan Weather and Climate. Section 1.2 reviews previous studies that have conducted evaluations of the meteorological inputs and hydrologic outputs of Land Data Assimilation Systems in the Central Asia region. Section 2 (Methods) describes the hydrologic modeling system, parameters and meteorological inputs, and model outputs. Section 3 (Results) presents comparisons of precipitation inputs, and comparisons of modeled snow estimates to remotely sensed snow observations. Finally, Section 4 describes an application of these data to the Afghanistan drought of 2018.

## 1.1 Afghanistan Weather and Climate

Central Asia, a region that includes Afghanistan, is water-scarce, receiving roughly 75% of its annual precipitation during November–April (Oki and Kanae, 2006). In Afghanistan, rainfall is highest in the northeast Hindu Kush Mountains and decreases toward the arid southwest Registan Desert (Fig. 1a). Temperature follows a similar pattern with cooler temperatures in the high elevation, wetter northeast, and warmer temperatures in the south and southwest (Fig. 1b). Regional precipitation is strongly influenced by the El Niño-Southern Oscillation (ENSO). La Niña conditions are associated with below average precipitation (FEWS NET, 2020b) and El Niño conditions are associated with above average precipitation (Barlow et al., 2016; Hoell et al., 2017; Rana et al., 2018; Hoell et al., 2018, 2020; FEWS NET, 2020a). Other factors with an important

influence on precipitation include orography, storm tracks, and the Madden–Julian oscillation
(Barlow et al., 2005; Nazemosadat and Ghaedamini, 2010; Hoell et al., 2018). The last several years
have experienced several ENSO events, with recent La Niña events in 2016-17, 2017-18, and 2020-
2022 (NOAA CPC ENSO Cold & Warm Episodes by Season, 2021) that corresponded to droughts
(FEWS NET, 2017b, 2018c, 2021).

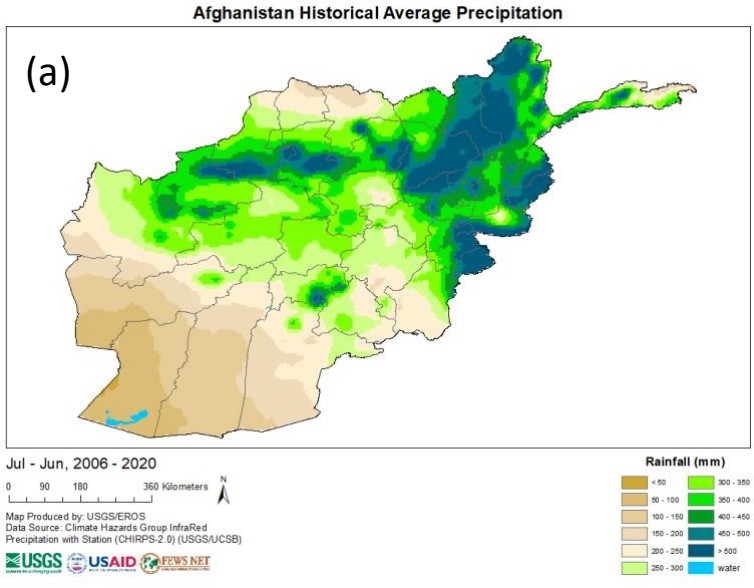

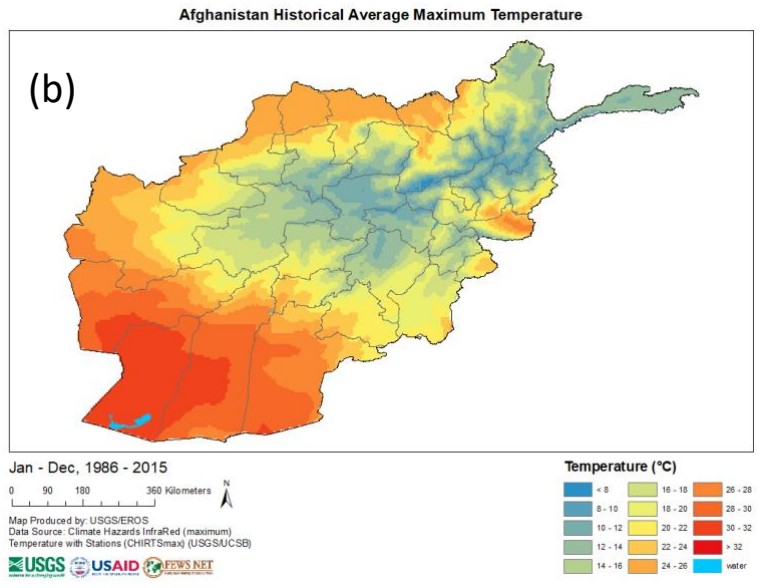

Figure 1. (a) Average annual precipitation in Afghanistan from 1991-2020, with overlayed province boundaries. (b) Average maximum monthly temperature from (1986-2015), overlayed with province boundaries. Map source USGS Knowledge Base (USGS Knowledge Base, 2021).

Despite Afghanistan's semi-arid climate, agriculture is an important sector, contributing 23% of its gross domestic product and employing 44% of the national labor force (CIA World Factbook). High mountain snowpack and snowmelt runoff are important for agricultural water supply. According to FEWS NET (2018b) snowmelt runoff is responsible for "providing over 80% of irrigation water used. The timing and duration of the snowmelt is a key factor in determining the volume of irrigation water and the length of time that it is available, as well as its availability for use in marginal areas that experience [variable] rainfall." Therefore, routine hydrologic monitoring, with a particular emphasis on snow, is critical for tracking agricultural conditions and provides early warning for food insecurity.

## 1.2 Hydrologic Data Availability and Uncertainty

Remote sensing and models are important inputs to climate services (Qamer et al., 2019). In the Central Asia region, and especially Afghanistan estimates of meteorological inputs, and model parameters have considerable uncertainty due to sparse in situ environmental observations. To address these challenges, the NASA High Mountain Asia project (https://www.himat.org/) has broadly aimed to explore the driving changes in hydrology as well as model validation and data assimilation, and water budget processes from the Himalayas in the south and east to the Hindu Kush in the west. These efforts and other studies of satellite derived rainfall informed the configuration and interpretation of the FLDAS Central Asia and global data streams.

The primary challenge to producing and evaluating hydrologic estimates is that sparse in situ precipitation observations lead to uncertainty in gridded, satellite-based precipitation estimates. Precipitation station observations are used for (a) bias correction of satellite estimates and (b) validation of gridded products. In terms of gridded dataset development, Hoell et al. (2015) describe how lack of station observations and complex topography in Afghanistan, Iraq, and Pakistan makes this issue particularly problematic. Barlow et al. (2016) also highlight the station availability across the region and how that influences uncertainties in the Global Precipitation Climatology Center (GPCC) version 6 (Schneider et al., 2017) dataset over Central Asia (Fig. 2a) and specifically Afghanistan over time (Fig. 2b).

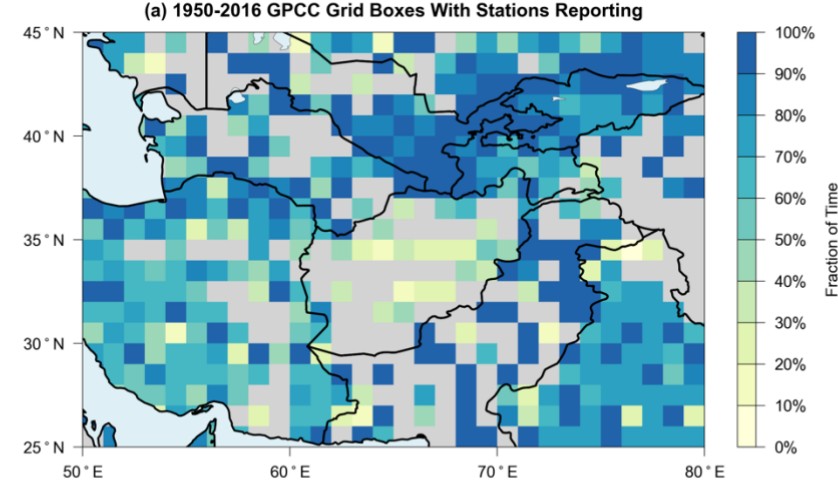

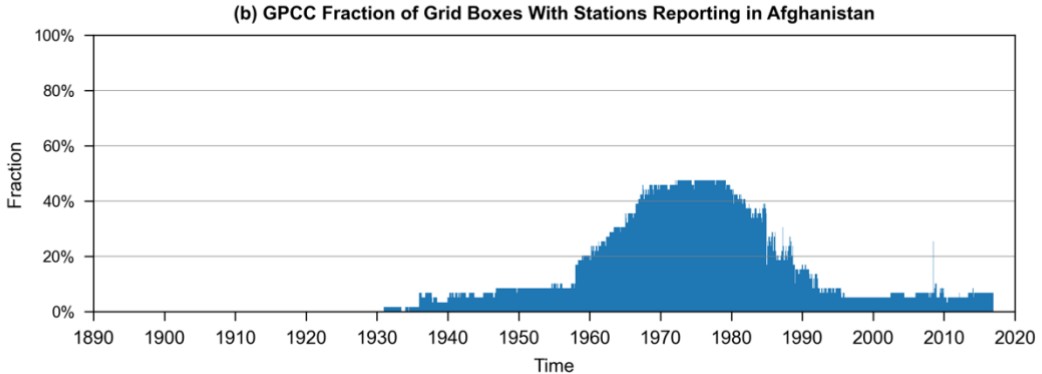

Figure 2. (a) Station data availability underlying the GPCC version 6 dataset, for the 1950–2016 period, on the 0.5°-resolution grid over Central Asia. (b) Fraction of gridcells with Number of stations used as input to the GPCC rainfall dataset in Afghanistan from 1932-2016.

In the absence of abundant in situ observations, one approach for remote sensing and model evaluation is to compare multiple input datasets and evaluate the water balance. Independent observations from the different components of the water balance (e.g., evapotranspiration, soil moisture, streamflow) help constrain estimates. We provide some background here and refer readers and data users to literature from the NASA High Mountain Asia project, specifically Yoon et al. (2019) and Ghatak et al. (2018), who explored similar configurations to the FLDAS system. This background allows the reader to appreciate the uncertainties in inputs, outputs and derived products and climate services over Afghanistan and the broader Central Asia region.

Meteorological forcing is known to be the primary source of uncertainty in land surface model simulations (Kato and Rodell, 2007). Thus, its evaluation is important to understand the quality of model inputs and outputs. For this reason, Ghatak et al. (2018) compare four unique precipitation data sources: daily Climate Hazards center Infrared Precipitation with Stations (CHIRPS) (Funk et

al., 2015), NOAA's Global Data Assimilation System (GDAS) (Derber et al., 1991), and two estimates from NASA's Modern Era Reanalysis for Research and Applications version 2 (MERRA-2) (Gelaro et al., 2017). They find that annual CHIRPS and GDAS precipitation estimates had similar bias and root mean squared error over Afghanistan with respect to APHRODITE (Asian Precipitation Highly Resolved Observational Data Integration Toward Evaluation) rain-gauge derived product (Yatagai et al., 2012). CHIRPS had a higher correlation with APHRODITE. Ghatak et al. (2018) further evaluated the quality of rainfall inputs based on the performance of evapotranspiration and other derived outputs. The authors caution that gridded precipitation estimates that have in situ inputs, like CHIRPS, may systematically underestimate precipitation in mountainous regions. We keep this consideration in mind when interpreting differences between FLDAS global and Central Asia data streams.

Yoon et al. (2019) compare precipitation estimates from 10 different products including APHRODITE, CHIRPS, GDAS, and MERRA-2, across a broad region of High Asia, including a portion of Afghanistan. They find that all datasets generally capture the spatial pattern of rainfall and that the products tend to agree more at high elevations, where it is unlikely there are station observations. Like Ghatak et al. (2018), they found CHIRPS and APHRODITE to have a lower average precipitation than GDAS, attributable to the incorporation of sparse gauge data.

In addition to precipitation, other meteorological inputs are important for accurate hydrologic estimates. Yoon et al. (2019) conducted an intercomparison of near surface air temperature estimates from three model analysis products (European Centre for Medium-Range Weather Forecasts (ECMWF; Molteni et al., 1996), GDAS, and MERRA-2). They noted a statistically significant upward trends in GDAS and ECMWF temperature, as well as consistently higher temperatures in MERRA-2. We see the same pattern when averaging across Afghanistan. Yoon et al. (2019) conclude that improvements in the meteorological boundary conditions would be needed to reduce the uncertainty in the terrestrial budget estimates. These sentiments are echoed in Qamer et al. (2019).

Despite known uncertainties, Schiemann et al. (2008) find that gridded precipitation estimates can qualitatively identify large scale spatial distribution of precipitation, seasonal cycles, and interannual variability (i.e., wet and dry years) across Central Asia. Long-term estimates of rainfall from satellite derived products, as well as derived historical time series from hydrologic modeling, can be used as a baseline of "observations," from which we can have a sense of relative conditions, i.e., anomalies and variability. When this historical record is harmonized with a routine monitoring system, current conditions can be placed in historical context. Anomaly-based representation of hydrologic extremes can provide confidence in modeled estimates that have the potential to influence agricultural, water resources and food security outcomes. For these reasons one of the requirements for FLDAS input is that there is a sufficiently long historical record for contextualizing estimates in terms of anomalies.

From a climate services perspective, the reliance on the representation of relatively wet and dry
conditions, as well as a "convergence of evidence" approach, provide useable information despite
the above-mentioned uncertainties. A convergence of evidence approach that draws on (quasi-)
independent sources of information is useful to understand actual conditions. For convergence of
Earth observations, hydrologic models can generate ensembles of historical, current, or future
estimates of snow, streamflow, soil moisture, and evapotranspiration, which can then be compared
to satellite derived estimates of surface water (e.g., McNally et al., 2019), soil moisture (e.g.,
McNally et al., 2016), vegetation conditions and evapotranspiration (e.g., Jung et al., 2019; Pervez
et al., 2021), snow cover (e.g., Arsenault et al., 2014), in situ streamflow (e.g. Jung et al., 2017) and
others. Hydrologic estimates can also be compared to outcomes in crop production (e.g., (e.g.,
McNally et al., 2015; Davenport et al., 2019; Shukla et al., 2020), and nutrition, health, and food
security (e.g., Grace and Davenport, 2021) to provide a qualitative understanding of both hydrologic
model performance and conditions on the ground. In this paper we provide an example for 2018
where drought conditions were associated with crisis levels of acute food insecurity over most of
Afghanistan (FEWS NET, 2018c).
To summarize, our experience and the literature have characterized uncertainties in available
meteorological forcing for the region. GDAS, CHIRPS, and MERRA-2 were chosen for the FLDAS
system based on our project requirements of (a) a sufficiently long historical record for
contextualizing estimates in terms of anomalies (b) low latency (< 1-month) for timely decision
support, (c) familiar to the FEWS NET user-community, and (d) prior evaluation by our team and
the broader community. We note here and describe in more detail later that the Integrated Multi-
satellite Retrievals for the Global Precipitation Mission (IMERG), a NASA precipitation product
(Huffman et al., 2020) also meets these requirements, since version 6 which was released in 2019
(after these studies and initial FLDAS configuration). We will a describe IMERG, GDAS, and
MERRA-2 comparison in the Results (Section 3).
**2 Methods**
**2.1 Land Surface Modeling System & Parameters**
A land surface model (LSM) can provide spatially and temporally continuous information about the
water and energy budgets of the land surface. This information is useful for food and water security
applications in places where in situ measurements of rainfall, soil moisture, snow and runoff are
sparse. This is particularly relevant in mountainous places like Afghanistan where heterogeneous
geography limits the representativeness of sparse in situ measurements. The FLDAS (McNally et
al., 2017) utilizes the  NASA's Land Information System Framework (LISF), which is composed of
a pre-processor, the Land surface Data Toolkit (Arsenault et al., 2018), the Land Information
System (Kumar et al., 2006; Peters-Lidard et al., 2007), and the Land Verification Toolkit (Kumar
et al., 2012). In this data descriptor we describe the two configurations of the FLDAS data streams
used for Central Asia food and water security applications. It uses the Noah 3.6 LSM (Chen et al.,
1996; Ek et al., 2003) for the two data streams (Fig. 3 and Table 1). The first data stream is global,
at ~1 month latency, and provides monthly average outputs on a 10-km grid from 1982-present. The
second data stream centered on Central Asia, ~1 day latency, provides daily average outputs at 1-km
from 2001-present.
One important feature, added by the NASA LISF software development team, is the radiation
correction described in Kumar et al. (2013), which improves the representation of snow dynamics
with respect to slope and aspect corrections on the downward solar radiation field. Another
noteworthy feature is the method of the Central Asia data stream restart (i.e., annual initialization
based on climatology), which was developed to address an issue of excessive inter-annual snow
accumulation found in the Noah LSM. First, a nine-year spin-up of the system was performed to
produce stable snow and soil moisture conditions. Next, the resulting model states were compared
with the Moderate Resolution Imaging Spectroradiometer (MODIS) Maximum Snow Extent data
originally computed by NOAA National Operational Hydrologic Remote Sensing Center (Greg Fall,
NOAA Operational Data Center, written communication., 2014). Then, the model-estimated
conditions were adjusted to produce a climatological model state for 1 October that is used to
initialize each year. This approach ensures that the 'water year,' beginning 1 October, is initialized
with a reasonable initial amount of snowpack. While this method does effectively manage excessive
inter-annual modeled snow accumulation, the user should be aware that using the climatological
model state will persist for ~1-2 months in the water and energy balance of the LSM until they are
superseded by "observed" meteorological inputs for the current water year. Preliminary work
indicates that this issue will be resolved in future updates.  In contrast, the global data stream does
not use this 1 October initialization procedure.
Although the two data stream specifications are largely the same, there are some differences related
to the input forcings, parameters and specifications (Table 1) and model spin-up procedures.

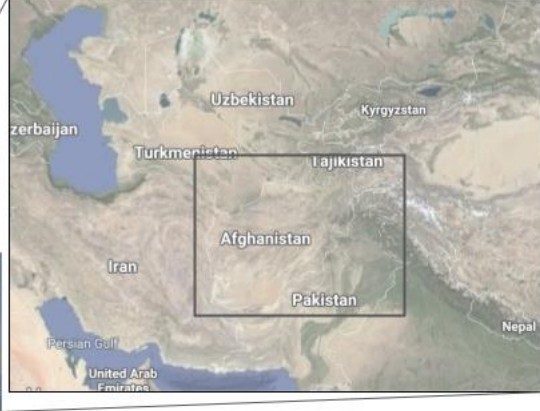

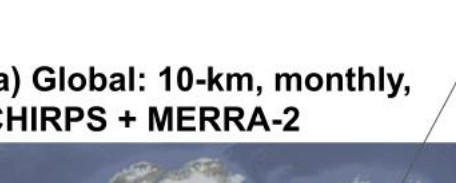

Figure 3. The FEWS NET Land Data Assimilation System (FLDAS) domains for (a) the global data
stream at 10-km spatial resolution and ~1 month latency for monthly averaged hydrologic estimates
and (b) the Central Asia data stream at 1-km spatial resolution and ~1 day latency for daily averaged
hydrologic estimates.

Table 1. FEWS NET Land Data Assimilation System (FLDAS) specifications for (A) global data
stream, 10-km monthly with CHIRPS+MERRA-2; and (B) Central Asia data stream, 1-km, daily
with GDAS.

|  | Global | Central Asia |
|---|---|---|
| Spatial Extent | 179.95°W- 179.95°E, 59.95°S-89.95°N | 30-100°E, 21-56°N |
| Landmask | Generated from MODIS using LISF-LDT, with MOD44w mask applied post-processing. | MOD44w (Carroll et al., 2017) |
| Landcover | IGBP landcover | IGBP landcover |
| Elevation | Shuttle Radar Topography Mission SRTM (NASA JPL, 2013) | SRTM |
| Albedo | National Centers for Environmental Prediction (NCEP) albedo (Csiszar | NCEP albedo & MODIS-based Max Snow Albedo |

| | and Gutman, 1999) & MODIS-based Max Snow Albedo (Barlage et al., 2005) | |
|---|---|---|
| | | |
| Vegetation Parameters | NCEP greenness fraction (Gutman and Ignatov, 1998) | NCEP greenness fraction |
| Non-Precipitation Meteorological Inputs | MERRA-2 | GDAS |
| Soil Texture | Food and Agricultural Organization (FAO) soil texture & properties (Reynolds et al., 2000) | FAO soil texture & properties |
| Precipitation Inputs | CHIRPS daily precipitation, downscaled to 6-hourly with LDT | GDAS 3-hourly precipitation |
| Specifications | Noah 3.6.1 | Noah 3.6.1 |
| Map Projection | Geographic Latitude-Longitude | Geographic Latitude-Longitude |
| Software Version | 7.2 | 7.3 |
| Spatial Resolution | 10-km | 1-km |
| Temporal Coverage | 1982-01-01 to present | 2000-10-01 to present |
| Model Timestep | 15-min timestep | 30-min timestep |
| Met. Forcing Heights | 2-m Air Temperature (Tair), 10-m Wind | 2-m Tair, 10-m Wind |
| Soil layers (meters) | 0-0.1; 0.1-0.4; 0.4-1.0; 1-2 | 0-0.1; 0.1-0.4; 0.4-1.0; 1-2 |
| Features | radiation correction | radiation correction |


The parameters and specifications listed in Table 1 are largely default settings defined by the Noah
LSM community (NCAR Research Applications Library, 2021). Ongoing research aims to identify
where model output performance can be improved with parameter updates. Evaluating parameter
updates had similar challenges as evaluating input forcing described in Section 1.2: without reliable
reference data it is difficult to determine a "best" input. For example, we have explored changing
soil parameters from FAO to International Soil Reference and Information Centre (ISRIC) SoilGrids
database (Hengl et al., 2017). This change did not result in improvements in streamflow statistics in
southern Africa, nor in soil moisture anomalies' ability to represent drought events. We expect
similar results in Afghanistan where, e.g., streamflow will be sensitive to a change in soil
parameters and the lack of referenced data to evaluate if there is an improvement. Moreover, our
model runs at 0.1 and 0.01 degrees may not fully exploit the added value of the 250m soil grids as
noted in Ellenburg et al. (2021) for a LISF application in East Africa.
Vegetation parameters are also potential sources of improvement whose importance to LDAS
hydrologic estimates has been highlighted (e.g., Miller et al., 2006). We have found the NCEP
estimates of green vegetation fraction (GVF) to be sufficient for this configuration of Noah 3.6. We
found that a time series of GVF derived from the Normalized Difference Vegetation Index (NDVI)
did not improve representation of droughts in eastern Africa. However, future FLDAS global and
Central Asia versions can be run with Noah-Multi parameterization (Noah-MP) (Niu et al., 2011)
which has multiple vegetation options and relies on either Leaf Area Index rather or GVF. This
model update is expected to open possibilities for choice of datasets to meet our application needs
and potentially improve representation of the water balance.

## 2.2 Meteorological Forcing Inputs

As previously discussed, precipitation is a critical input to land surface models. The lower-latency
Central Asia data stream is a daily product, forced with GDAS (Derber et al., 1991) 3-hourly
precipitation, which is available from 2001to present at <1-day latency. This dataset was chosen
because of its latency. The global data stream is driven by the daily CHIRPS product (Funk et al.,
2015), which is available from 1981 to present at ~ 5-day latency for CHIRPS Preliminary and ~1.5-
month latency for CHIRPS Final. The CHIRPS products were chosen as inputs because of their
proven performance in the literature, which has made it the "gold standard" for food and water
security monitoring by organizations like FEWS NET, the World Food Program, and others who
need up-to-date estimates and a 40+ year historical record. As mentioned earlier, lack of rainfall
stations for bias correction of satellite-derived estimates and evaluation poses a major challenge.
However, we find that the GDAS rainfall product and the CHIRPS rainfall product are adequate for
routine monitoring and, along with additional sources of remote sensed information, are important
for convergence of evidence when making a best estimate at land surface states and fluxes.
Before the daily CHIRPS rainfall data can be used as input to the FLDAS models, the daily
precipitation is pre-processed to a sub-daily timestep, using the LDT component of the LISF
software. LDT temporally disaggregates the daily CHIRPS rainfall using an approach similar to the
North American LDAS precipitation temporal downscaling (Cosgrove et al., 2003). For this
approach, we use a finer timescale MERRA-2 precipitation timescale as a reference dataset to
represent an accurate diurnal cycle. We note that this step in our methodology facilitates the solving
of FLDAS water and energy balances at a sub-daily timestep. However, for Central Asia we do not
have sufficient reference data available to assess the importance of sub-daily precipitation
distribution, as was demonstrated by Sarmiento et al. (2021) for the United States where adequate
reference data are available.  For spatial downscaling, coarser scale meteorological forcings are
spatially disaggregated to the output resolution (0.01, and 0.1 degree for Central Asia and global,
respectively) in the LISF using bilinear interpolation.
The FLDAS models require additional meteorological inputs, including air temperature, humidity,
radiation, and wind. The lower-latency Central Asia data stream uses GDAS 3-hourly
meteorological inputs available from 2001-present at <1-day latency. For a longer historical record,
the global data stream uses MERRA-2 (Gelaro et al., 2017) (1979-present) 1-hourly products with a
two-week latency. Over the Afghanistan domain GDAS temperature has an upward trend, whereas
MERRA-2 is consistently warmer before 2010. We find that GDAS and MERRA-2 temperature
estimates are of similar magnitude during 2011-2020. Similar results were noted by Yoon et al.
(2019) who found an upward trend in GDAS temperature, as well as consistently higher
temperatures in MERRA-2 across a broad High Asia domain.
**2.3 Model Evaluation Statistics and Comparison Data**
In addition to guidance from previous studies (Section 1.2), we assessed the quality of our modeling
outputs by conducting comparisons between (1) FLDAS satellite rainfall inputs and other satellite
precipitation estimates, and (2) model estimated snow cover fraction and satellite derived snow
cover fraction estimates.
For the precipitation analysis, we compare CHIRPS and GDAS inputs to the Integrated Multi-
satellite Retrievals for the Global Precipitation Mission (IMERG), a NASA precipitation product
that integrates passive microwave and infrared satellite data with surface station observations
(Huffman et al., 2020). The IMERG Final Run precipitation product, available at ~ 2-month latency
(thus not suitable for our monitoring applications) has been used in numerous verification studies,
including studies over Africa (Dezfuli et al., 2017), South America (Gadelha et al., 2019; Manz et
al., 2017), and the mid-Atlantic region of the United States (Tan et al., 2016). These studies
demonstrated that IMERG Final Run was able to capture large spatial patterns and seasonal and
interannual patterns of rainfall. However, fewer studies have explored the performance of the lower
latency IMERG Late Run (doi: 10.5067/GPM/IMERGDL/DAY/06) product that we use here.
Kirshbaum et al. (2016) include a qualitative comparison for CHIRPS Final and IMERG Late Run
for the Southern Africa start-of-season 2015. IMERG Late Run appears to perform similarly to the
1.5-month latency CHIRPS Final and outperform the 1-day latency NOAA Rainfall Estimate
version 2 (RFE2) product (Xie and Arkin, 1996). Differences in the daily rainfall distribution
patterns between IMERG Final Run and CHIRPS Final have also been shown to affect the resulting
hydrological modeled output in simulations done using the NASA LISF (Sarmiento et al., 2021).
For the snow cover fraction (SCF) analysis, we compare the global and Central Asia data streams
with the MODIS daily SCF product, MOD10A1 Collection 6 (Hall and Riggs, 2016). MOD10A1
data are available at 500-m spatial resolution from February 2000 to the present. SCF is generated
using the Normalized Difference Snow Index (NDSI) and additional filters to reduce error and flag
uncertainty. Routine qualitative comparisons, which can be viewed on the NASA LISF FEWS NET
project website, generally show agreement between the model and MODIS SCF, as well as
occurrence of cloud cover (https://ldas.gsfc.nasa.gov/fldas/models/central-asia). Following
Arsenault et al. (2014), we aggregated pixels to 0.01 degree to reduce error related to sensor viewing
angles and gridding artifacts. For this analysis, using MODIS SCF as "truth," we determined True
Positives (TP), True Negatives (TN), False Negatives (FN) and False Positives (FP). We then
computed probability of detection (POD) where POD = (TP/(TP + FN)) and False Alarm Rate
(FAR) where FAR = (FP/(FP + TN)). We computed these for the total area of Afghanistan (60-76E,
28-39N), as well as by basin (Fig. 4). This paper does not compare modeled snow water equivalent
(SWE) to independent snow observations because, as noted by Yoon et al. (2019), direct evaluation
of snow mass and SWE)is difficult over Central Asia due to poor coverage of accurate snow
observations. We follow the Yoon et al. (2019) recommendation to conduct quantitative SCF
comparisons and provide qualitative SWE analysis in Applications, Section 4.
In addition to rainfall and snow comparisons, we conducted monthly pixel-wise comparison of
Central Asia and the global run's estimates of evapotranspiration (ET) and soil moisture versus
Operational Simplified Surface Energy Balance (SSEBop, (Senay et al., 2013)). ET and Soil
Moisture Active Passive (SMAP) Level 3 (Entekhabi et al., 2010, 2016) using the Normalized
Information Contribution (NIC) metric following Sarmiento et al., (2021). The analysis was
performed for the period 2016-2021 to match the SMAP record. The NIC metric first computes
anomaly correlations between the model runs and the reference dataset and then computes the
difference between the performance of each model run using a scale of -1 to +1 to highlight if the
global or Central Asia data stream performs better with respect to the reference. To make the
comparisons, the reference datasets (SMAP and SSEBop) were re-gridded to match the grid spacing
and locations of the experiment model outputs.

## 379  3 Results

## 380  3.1 Gridded Rainfall Comparison

We have two data streams for Central Asia applications with different precipitation inputs: 1) the
global data stream with CHIRPS precipitation at 10-km spatial resolution provides a long-term
consistent data record; and 2) the Central Asia data stream with GDAS precipitation at 1-km
provides near real time, finer spatial resolution updates. These two data streams have their
respective errors and allow data users to apply a convergence of evidence approach for food and
water security applications. This section presents a comparison of the GDAS, and CHIRPS
precipitation inputs used for the Central Asia and global data streams, respectively. We also include
IMERG Late Run for comparison as a high quality, low latency product. Future work may
incorporate the IMERG Late Run precipitation inputs into FLDAS simulations. We also include
MERRA-2 precipitation for comparison. Pair-wise correlations are shown in Table 2. CHIRPS
Final, IMERG Late Run and GDAS (R $\geq$ 0.90) are well correlated in terms of average daily
precipitation (mm/day) at the monthly and annual (i.e., water year) timestep. MERRA-2 correlations
with these datasets are lower at the monthly (0.75 $\leq$ R $\leq$ 0.81) and water year (0.64 $\leq$ R $\leq$ 0.69)
timesteps. Fig. 4 shows the time series of the precipitation products for their overlapping period of
record (2001-2020), which illustrates how they vary in time, and shows some general patterns in
terms of relative precipitation in mm: GDAS (blue) and IMERG Late Run (purple) tend to have the
highest precipitation totals, CHIRPS (green) has lower precipitation but is higher than MERRA-2
(yellow) which tends to have the lowest precipitation, until 2019 when it is notably higher than the
other products.

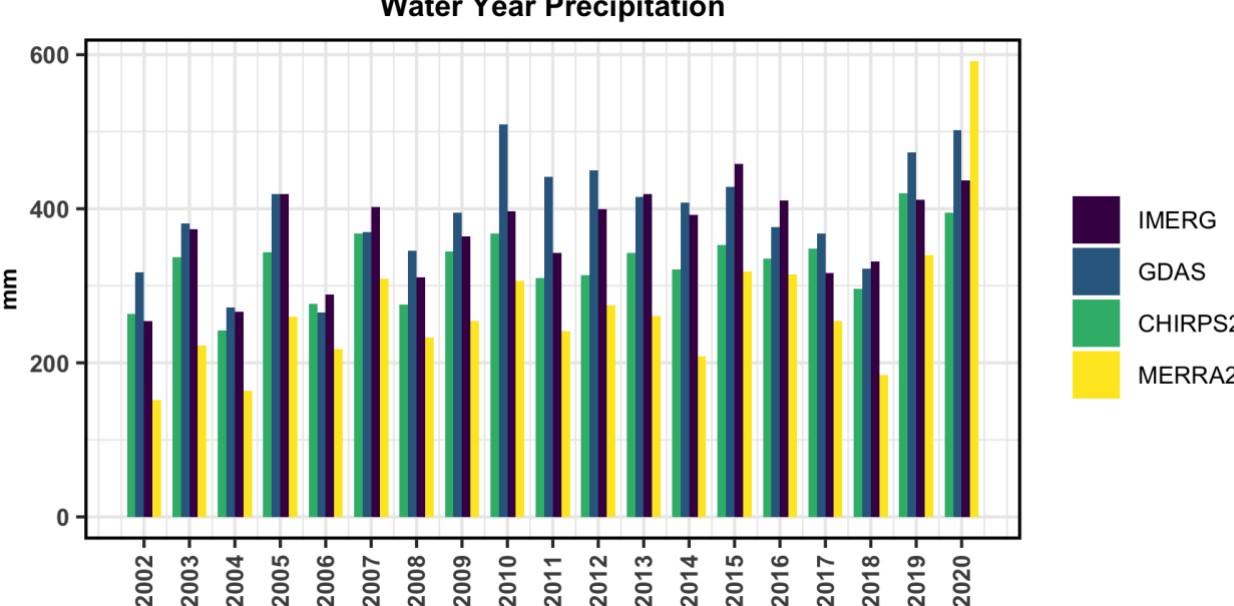

Figure 4. Afghanistan water year precipitation for CHIRPS, GDAS, IMERG Late Run, and
MERRA-2.
Table 2. Afghanistan spatial average Spearman Rank Correlation (R) of monthly (water year)
precipitation 2001-2020

|  | GDAS | CHIRPS Final | IMERG Late Run |
|---|---|---|---|
| GDAS | x | - | - |
| CHIRPS Final | 0.91 (0.92) | x | - |
| IMERG Late Run | 0.91 (0.89) | 0.92 (0.90) | x |
| MERRA-2 | 0.75 (0.64) | 0.78 (0.68) | 0.81(0.69) |


**3.2 Remotely Sensed and Modeled Snow comparisons**
The estimation of snow is important for Afghanistan and Central Asia because it is a critical
contributor to water resources and irrigated agriculture. We compared average SCF (Fig. 6a), POD,
and FAR statistics (Fig. 6b) relative to MODIS SCF over eight hydrologic basins in Afghanistan.

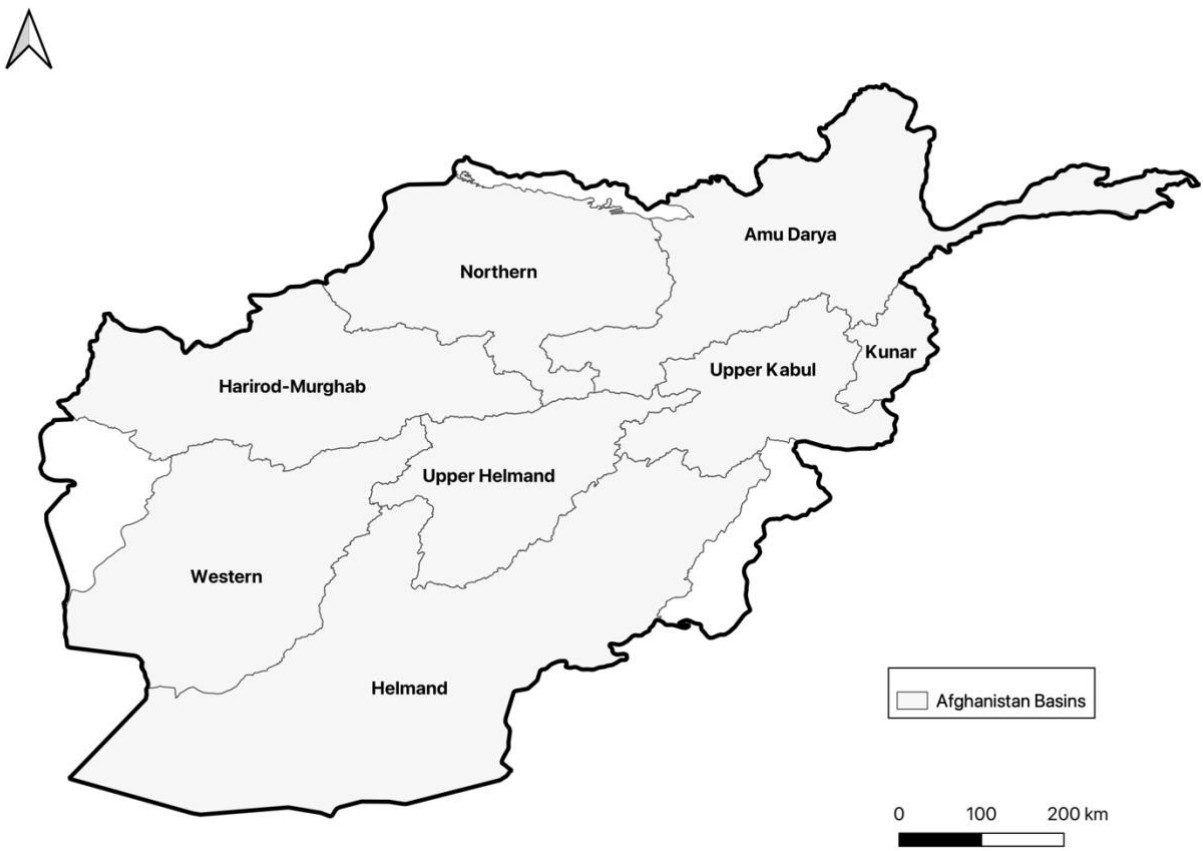

Figure 5. Hydrologic basins used in the analysis of categorical statistics for snow covered fraction.

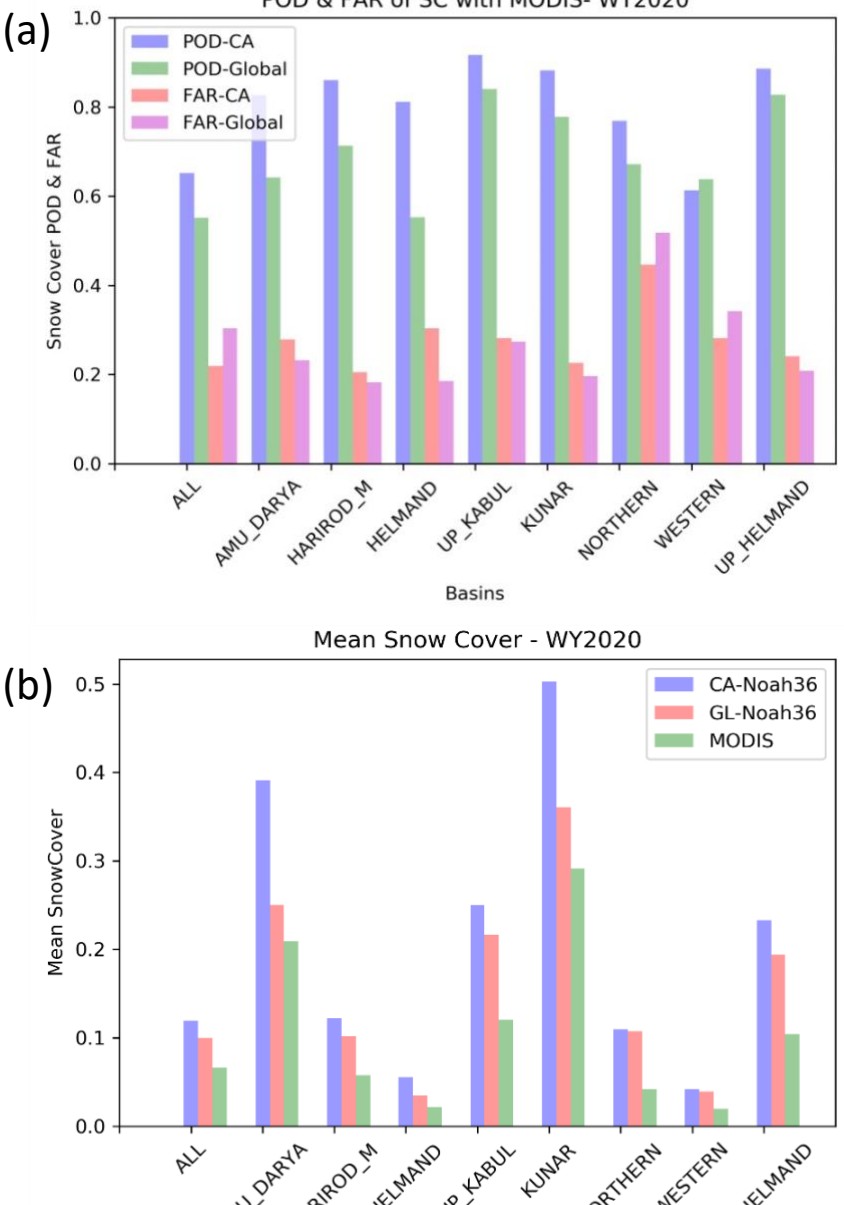

Figure 6. (a) Mean snow cover fraction for the entire area and by hydrologic basin for MODIS
Snow Cover Fraction (SCF), Central Asia (CA) and global (GL) data streams for water year 2020.
(b) Probability of Detection (POD) of snow presence, and False Alarm Rate (FAR) for the Central
Asia (CA) and global data streams relative to the MODIS SCF for water year 2020.

Overall, both model runs estimate greater average SCF than the MODIS SCF product. The Central
Asia data stream has consistently higher average snow cover for all basins compared to MODIS
SCF estimates and the global data stream. Perhaps not surprisingly that the Central Asia data stream
performs consistently better in POD (by basin = ~80%) except for the Western Basin. Similarly, the
FAR of the Central Asia data stream is higher where POD is higher except for the Northern Basin.
The difference in statistics may be related to the different forcing inputs or the higher spatial
resolution of the Central Asia data stream. Kumar et al. (2013) note that higher spatial resolution
was important for snow dominated basins.
In addition to precipitation and snow cover comparisons we conducted comparisons with remotely
sensed soil moisture and ET (not shown). We found that in general, GDAS derived estimates of ET
consistently performed better over Afghanistan in terms of pixel-wise anomaly correlation and NIC
with SSEBop ET. Meanwhile, neither modeled estimate of soil moisture consistently outperformed
the other with respect to SMAP. The ET results lend some support to the quality of the Central Asia
data stream estimates. However, the lack of signal in the soil moisture comparisons suggests that
more careful analysis of the model and remote sensing errors is required before drawing conclusions
regarding which data stream is "best."
**3.3 Discussion of results compared to previous studies**
Despite the lack of ground-based observations, our analysis shows that the remotely sensed
estimates and the models have good correspondence with other sources of evidence in terms of
seasonal timing and performance. This provides analysts with confidence when using the FLDAS
snow estimates, in tandem with other sources, as an input to food security assessments. Our
approach is supported by other studies that have explored the challenges of evaluating hydrologic
estimates over the region (Immerzeel et al., 2015; Ghatak et al., 2018; Yoon et al., 2019; Qamer et
al., 2019) .
Yoon et al. (2019) show that their LSM ensembles of SCF have an average POD of 72% and FAR
of 36%, which is within the range of our POD and FAR statistics (60-80% POD; 20-40% FAR)
compared to MODIS SCF.  The categorical statistics indicate that Central Asia (GDAS) tends to
have both a higher probability of detection and false alarm rate, indicating higher averages than
MODIS SCF and global (CHIRPS).
With respect to the soil moisture and ET comparisons, we found that the Central Asia data stream
estimates of ET were better correlated with SSEBop ET, but neither data stream was consistently
better correlated with SMAP. These differences could be a function of non-precipitation differences,
or higher spatial resolution. Ghatak et al. (2018) also found that the choice of reference dataset (with
its own characteristics and errors) was an important factor.

In general, given the lack of clarity on "best" FLDAS data stream, the convergence of evidence
approach allows us to consult both data streams, leveraging the longer time series of CHIRPS and
the lower latency of GDAS.
**3.4 Limitations and Future Developments**
Given the need for multiple data streams for convergence of evidence, we have summarized the pros
and cons of the Central Asia and global data streams in Table 3.
Table 3. Pros and cons of the two data streams

| | Central Asia: Noah 3.6 with GDAS (2000-present) | Global: Noah 3.6 with CHIRPS+MERRA-2 (1982-present) |
|---|---|---|
| Pros | 1-km | less computationally intensive |
| | 1-day latency, daily timestep | longer time record |
| | Snow estimates available in USGS Early Warning eXplorer https://earlywarning.usgs.gov/fews/ewx/ | CHIRPS & MERRA-2 forcing spatial resolution does not change over time (stable climatology) |
| | | Water and Energy balance available in NASA GIOVANNI https://giovanni.gsfc.nasa.gov/giovanni/; Google Earth Engine https://developers.google.com/earth-engine/datasets/tags/fldas; Climate Engine https://climateengine.com/ |
| Cons | more computationally intensive | lower resolution (10-km) |
| | shorter time record | ~30-day latency |
| | GDAS forcing resolution changes over time (unstable climatology) (NOAA NCEP https://www.emc.ncep.noaa.gov/gmb/STATS/html/model_changes.html) | not publicly available at daily timestep |
| | large data volume, difficult to move | |


IMERG version 6 was released in 2019 and includes rainfall estimates processed back to 2000. Prior
to this change we had found encouraging results when comparing the onset of rainy season using
both IMERG Late Run and CHIRPS (Kirschbaum et al., 2016). However, at that time the period of
record was a limitation for computing anomalies. We now have an adequate period of record, and
IMERG Late Run is planned to be part of the upcoming FLDAS global and FLDAS Central Asia
releases. We are also encouraged by the quality of IMERG at the daily timestep when compared to
CHIRPS over the United States where accurate reference data are available (Sarmiento et al., 2021).
In addition to IMERG other promising rainfall datasets are in development. Ma et al. (2020) have
developed the AIMERG dataset that combines IMERG Final Run with the APHRODITE rain-gauge
derived product (Yatagai et al., 2012).  Another promising dataset is CHIMES (Funk et al., 2022), a
blend of CHIRPS and IMERG, whose developers have been exploring the strengths and limitations
of these two datasets and their fusion to produce an optimal product.
With respect to other FLDAS developments, FLDAS global and Central Asia are planned to be
transition to Noah-MP. This will allow for improved representation of snowpack and groundwater.
This will also necessitate the use of different parameters, e.g., leaf area index, as well as the
potential to explore different parameter sets like ISRIC soils.  In the meantime, multi-forcing and
multi-model ensembles, and convergence of evidence with other remotely sensed data and field
reports, are a viable approach for providing hydrologic estimates for various applications.
**4 Applications**
These data from global and Central Asia data streams are routinely used in several FEWS NET
information products listed in Table 4. NOAA's Climate Prediction Center (CPC) International
Desks provide a weekly briefing on the past week's weather conditions and 1– 2-week forecasts for
FEWS NET regions of interest, including Central Asia. There is also a monthly FEWS NET
Seasonal Monitor and a monthly Seasonal Forecast Review for which these data provide
information on the current state of the snowpack, soil moisture, and runoff. These "observed
conditions" can then be qualitatively combined with forecasts 1 week to many months in the future
to assess potential hydro-meteorological hazards. To demonstrate the role of these data in the early
warning process, at different points in the season, we provide an example of the 2017-2018 wet
season in Afghanistan during a La Niña event that contributed to drought.
Table 4. Routine Applications of FLDAS Central Asia's Afghanistan hydrologic data.

| Routine application of these data | Weblink to updates | Notes |
|---|---|---|
| FEWS NET Global Weather Hazards | https://fews.net/global/global-weather-hazards/ | shapefiles https://ftp.cpc.ncep.noaa.gov/fews/weather_hazards/ |

| | | |
|---|---|---|
| Summary produced by NOAA CPC | https://www.cpc.ncep.noaa.gov/products/international/index.shtml | |
| Seasonal Monitor | https://earlywarning.usgs.gov/fews/afghanistan/seasonal-monitor | Updated near the middle of each month from October - May, the wet season. |
| FEWS NET Food Security Outlook Brief | https://fews.net/central-asia/afghanistan | Information on snow or other hydrology included if applicable |
| Crop Monitor for Early Warning | https://cropmonitor.org/index.php/cmreports/early-warning-report/ | Information on early warning and crop conditions |


## 4.1 Snow Monitoring & Seasonal Outlooks

As previously mentioned, and as shown in Fig. 7, Afghanistan and the broader region is strongly
influenced by La Niña, which tends to increase the likelihood of below average precipitation.
Depending on this and antecedent conditions there in an increased likelihood of below average
snowpack, reduce springtime streamflow and flood risk, reduce summer irrigation water
availability, and crop yield losses.

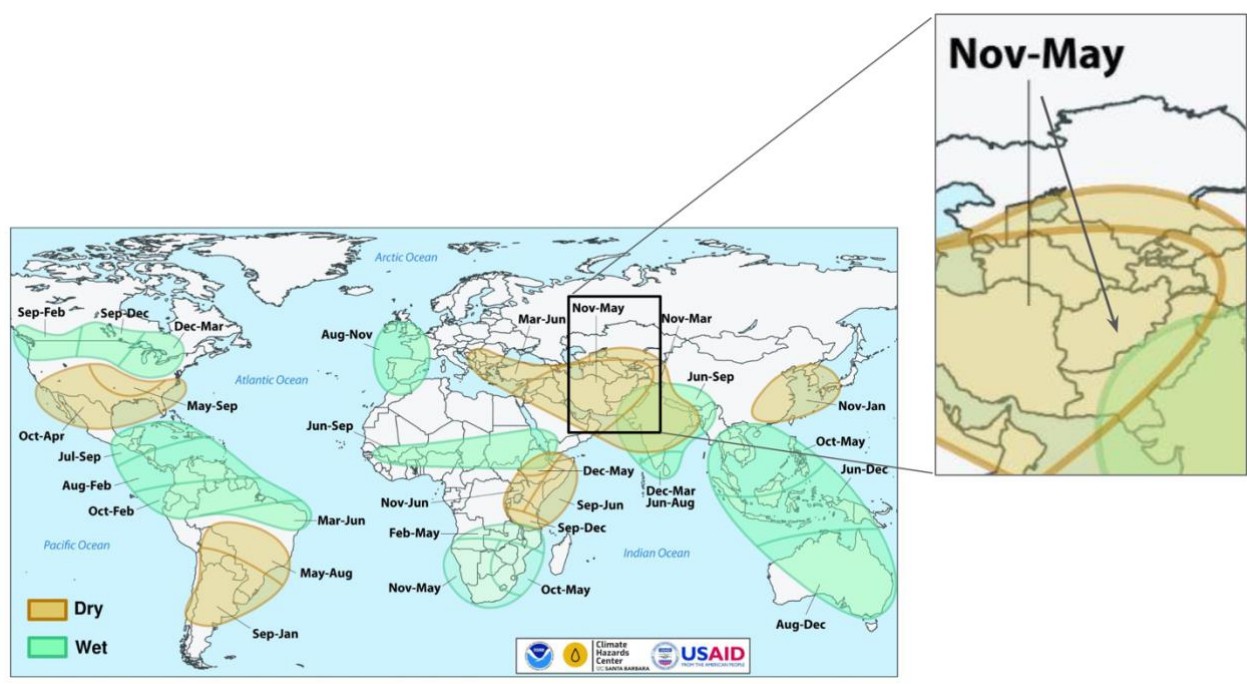


Figure 7. Timing of wet and dry conditions related to La Niña. Increased likelihood of dry
conditions from November-May for Afghanistan during La Niña events.
A La Niña Watch was issued by NOAA in September 2017 (NOAA, 2017). The FEWS NET
October 2017 Food Security Outlook (FEWS NET, 2017a) stated that La Niña conditions were
expected throughout the northern hemisphere fall and winter and that below-average precipitation
was likely over much of Central Asia, including Afghanistan, during the 2017-2018 wet season.
With the expectation of below average precipitation coupled with above average temperatures,
FEWS NET anticipated that snowpack would most likely be below average. In the context of food
security outcomes, it was assumed that areas planted with winter wheat were likely to be less than
usual, reducing land preparation activities and associated demand for labor. Two provinces of
particular concern were Daykundi and Wardak (Fig. 8a brown borders), both located in the
Helmand River Basin (Fig. 8a; gray shading).  Precipitation deficits in these provinces would lead to
poor rangeland resources and pasture availability and would likely result in decreased livestock
productivity and milk production through May. However, given that October was the start of the wet
season, there remained a large spread of possible outcomes: spatial and temporal rainfall
distributions, and snowpack totals necessitating routine updates to assumptions.
Monitoring continued during the wet season, tracking observations from remote sensing, models,
and field reports as well as forecasts across timescales. This information was used to regularly
update expectations of end of season outcomes. Using the FLDAS Central Asia data stream, a
December 21, 2017, NOAA CPC Weather Hazards Brief reported that parts of northern and central
Afghanistan remained atypically snow free, and north-eastern high elevation areas exhibited SWE
deficits. SWE is a commonly used measurement of the amount of liquid water contained within the
snowpack, and an indicator of the amount of water that will be released from the snowpack when it
melts. By January 17, 2018, an abnormal dryness polygon was placed over northeastern Afghanistan
and the central highlands, based on below-average SWE values from the FLDAS Central Asia
estimates. Abnormal dryness is defined for an area that has registered cumulative 4-week
precipitation and soil moisture ranking less than the 30th percentile, with a Standardized
Precipitation Index (SPI) of 0.4 standard deviation below the average. In addition, it is required that
forecasts indicate below-average precipitation (less than 80% of normal) for that area during the 1-
week outlook period. By late February 2018, precipitation deficits and related SWE (Fig. 9)
increased and met the criteria for "drought" (Fig. 8b). Drought is defined as an area that has
previously been defined as "Abnormal Dryness" and has continued to register seasonal precipitation
and soil moisture deficits since the beginning of the rainfall season. Specifically, an eight-week
cumulative precipitation, soil moisture, and runoff below the 20th percentile rank, and an SPI of 0.8
standard deviation below the average are classification guidelines.

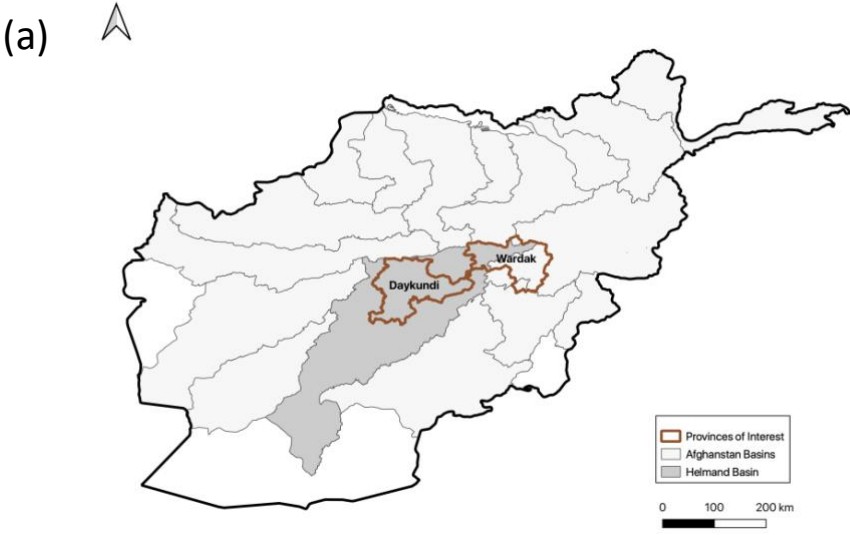

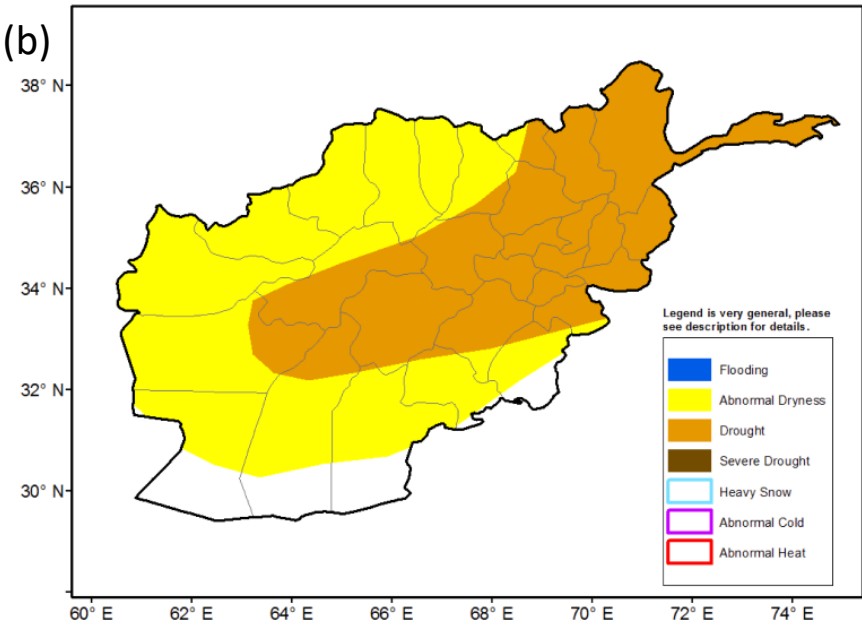

Figure 8. (a) Map showing hydrological basins, with Helmand Basin in darker gray and location of
Daykundi and Wardak provinces (outlined in red) where food security conditions were of particular
concern, (b) NOAA CPC Afghanistan Hazards Report for February 22-28, 2018 (CPC NOAA,
2018) showing widespread abnormal dryness and drought, defined by 90-day precipitation deficits
and extremely low snow water equivalent.

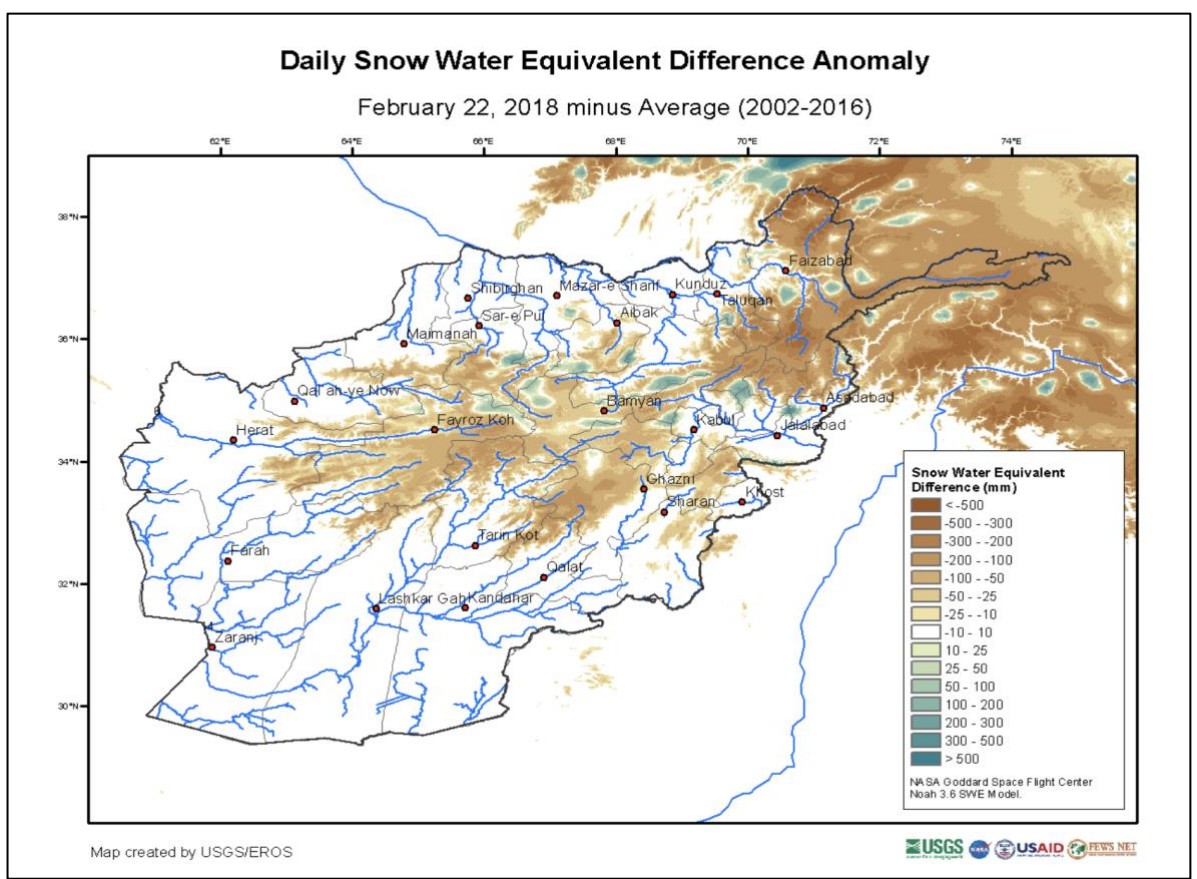

Figure 9. FLDAS Central Asia snow water equivalent (SWE) estimates for February 22, 2018.
SWE deficits of 300-mm were widespread at this time.
The February 2018 Food Security Outlook (FEWS NET, 2018b) provided the following updates,
based on the CPC Hazards Reports and Seasonal Monitors: "Snow accumulation and cumulative
precipitation were well below average for the season through February 2018, with some basins at or
near record low snowpack, with data since 2002….These factors will likely have an adverse impact
on staple production in marginal irrigated areas and in many rainfed areas. [Moreover, with]
forecasts for above-average temperatures during the spring and summer, rangeland conditions are
expected to be poor during the period of analysis through September 2018. This could have an
adverse impact on pastoralists and agro-pastoralists, particularly in areas where livestock
movements are limited by conflict." The Crop Monitor for Early Warning reports for February and
March 2018 (GEOGLAM, 2018a, b) also cited reduced snowpack in Afghanistan and the negative
impacts on winter wheat crops as well as irrigation water availability in the Spring. The story was
also highlighted in NASA Earth Observatory March 2018 "Record Low Snowpack in Afghanistan"
(NASA Earth Observatory, 2018).

The USGS Early Warning eXplorer (EWX) (Shukla et al., 2021) allows analysts to look at maps
and time series for a variety of variables and specific provinces and river basins. Plots from EWX in
Fig. 10 show below average precipitation for provinces in the Helmand Basin for January and
February. CHIRPS cumulative rainfall for 2017-18 versus the 18-year average for Day Kundi (a.k.a.
Daykundi) Province showed near average conditions until December. From January, cumulative
rainfall remained below the 2000-2018 average throughout the rest of the season ending in May; the
same pattern occurred in nearby Uruzgan Province. In neighboring Maydan Wardak (a.k.a Wardak)
Province, below average conditions were experienced in January and February, but cumulative
rainfall recovered in March to remain slightly above average. Day Kundi (Fig. 10b) and Wardak
(Fig. 10c) are provinces located in the upper reaches of the Helmand Basin. Fig. 10c shows SWE
averaged across the entire Helmand basin. The gray shading indicates the range of the minimum and
maximum values, and the dashed blue line is the average. Initial snow conditions start above
average until December, after which SWE deficits are near record low values through the beginning
of February, and then persist at below-average levels.

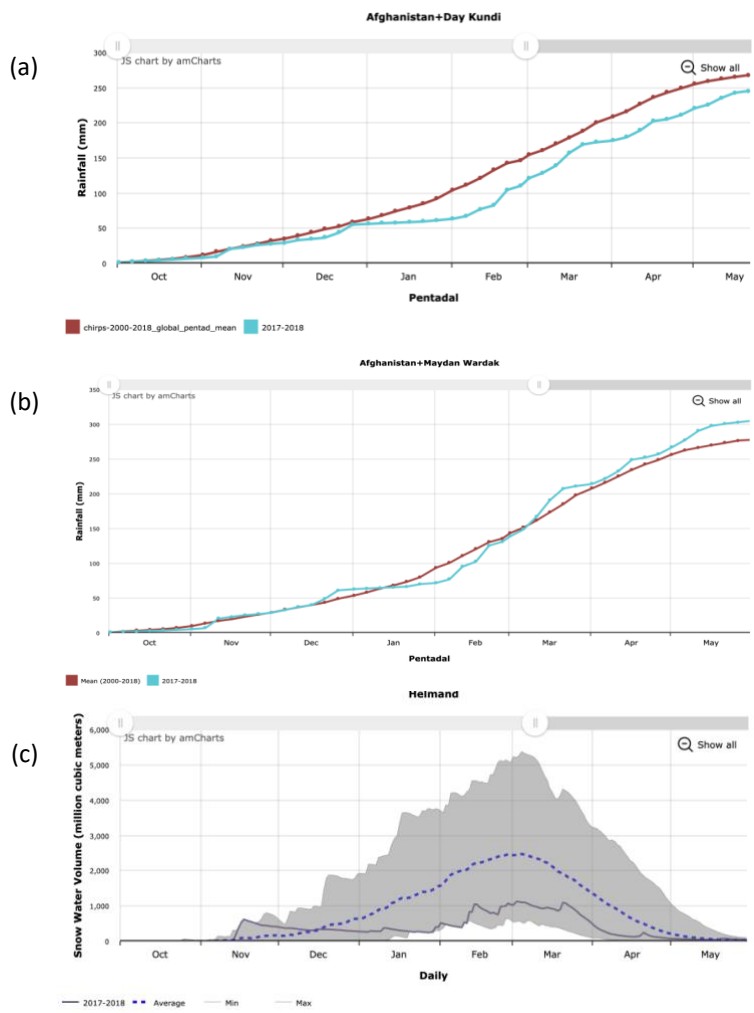


Figure 10. (a) CHIRPS cumulative rainfall for 2017-18 versus average conditions for Daykundi Province. (b) CHIRPS cumulative rainfall for 2017-18 versus average conditions for Maydan Wardak Province (c) Helmand Basin SWE from the FLDAS Central Asia data stream. The grey shading indicates the range of the minimum and maximum values, dashed blue line is the average, and black line is 2017-18. Figures from USGS EWX (https://earlywarning.usgs.gov/fews/ewx/).

By the end of the season in April 2018, FEWS NET (2018c) concluded that "below-average precipitation throughout most of the country during the October 2017 – May 2018 wet season has led to very low snowpack ...Low irrigation water availability is likely to have an adverse impact on yields for winter wheat and other ...barley, maize, and others.. particularly in downstream areas in regions with limited rainfall. ...The poor performance of the wet season and above average temperatures... exacerbated dry rangeland conditions in many areas, particularly in ...Sari Pul, [and surrounding] ...provinces. Pastoralists and agropastoralists in these areas will likely attempt to migrate to areas with better pasture and water availability or sell livestock at below-average prices." At the same time, UNICEF (2018) reported in April 2018 that among "the [drought] affected provinces, Baghis, Bamyan, Daykundi, Ghor, Helmand, ... and Uruzgan are of critical priority for nutrition and water, sanitation and hygiene assistance."

Several months after a season has ended, and harvest is complete, more statistics become available for further verification of the drought outcomes. The FEWS NET October 2018 Food Security Outlook (2018a) reported that the 2017-18 drought had significant negative impacts on rainfed wheat production and livestock pasture and body conditions across the country. Reporting statistics from the Afghanistan Ministry of Agriculture, Irrigation, and Livestock, the total wheat production for the 2017-18 season was about 20% below average, where irrigated agriculture performed about average. However, rainfed agricultural production was only about 50% of average, most severely affecting households in Badakhshan, Badhis, and Daykundi provinces. In these locations dry conditions, conflict, poor incomes, and depleted assets were expected to continue to face emergency food insecurity through May 2019.

## 5. Data Availability

The Central Asia data described in this manuscript can be accessed at the NASA GES DISC repository under data doi 10.5067/VQ4CD3Y9YC0R. The data citation is the following:

Jacob, Jossy and Slinski, Kimberly (NASA/GSFC/HSL) (2021), FLDAS Noah Land Surface Model L4 Central Asia Daily 0.01 x 0.01 degree, Greenbelt, MD, USA, Goddard Earth Sciences Data and Information Services Center (GES DISC) 10.5067/VQ4CD3Y9YC0R

The global data described in this manuscript can be accessed at the NASA GES DISC repository under data doi 10.5067/5NHC22T9375G. The data citation is the following:

McNally, Amy. NASA/GSFC/HSL (2018), FLDAS Noah Land Surface Model L4 Global Monthly
0.1 x 0.1 degree (MERRA-2 and CHIRPS), Greenbelt, MD, USA, Goddard Earth Sciences Data and
Information Services Center (GES DISC), 10.5067/5NHC22T9375G
Currently the USGS EROS Center provides images from these data:
https://earlywarning.usgs.gov/fews/search/Asia/Central%20Asia, as well as an interactive data
viewer, the USGS EWX (https://earlywarning.usgs.gov/fews/ewx/).
**6. Code availability**
The NASA Land Information System Framework (LISF) is publicly available and an open-source
software. The software and technical support are available at https://github.com/NASA-LIS/LISF.
**7. Conclusion**
This paper describes a comprehensive hydrologic analysis system for food security monitoring in
Central Asia, with analysis focusing on Afghanistan. While these data are tailored to specific needs,
they are also applicable to other climate services and research. Our intent is to provide the reader
with information regarding the configuration and specification of both the current global and Central
Asia data streams.  These data are publicly available and available at near-real time for food security
decision support. Note that, as an on-going initiative, FLDAS model version and parameters are
routinely updated, and the user should consult the version updates provided by the NASA Goddard
Earth Science Data and Information Services Center (GES DISC) data provider and documentation
on USGS Early Warning website. For example, efforts are currently underway to upgrade to the
Noah-MP (Niu et al., 2011) land surface model, which requires some changes in parameters for
snow, glaciers and groundwater. This, and future changes, can be informed by the strengths and
weaknesses of the data stream configurations that we have discussed in this paper.
This paper also provides model-model and model-remote sensing comparisons as well as a review
of other research that highlights the challenges of quantitative evaluation of models and remote
sensing in this region. A key challenge to hydrologic modeling is the considerable uncertainty in the
meteorological forcing available for this region, particularly precipitation. Advancements in remote
sensing and modeling should help reduce these uncertainties. In addition, the current land surface
modeling reflects natural conditions, i.e., they do not include representation of anthropogenic effects
such as human water abstractions (e.g., dams for flood control or irrigation, water diversions,
groundwater pumping) or land application of abstracted water (i.e., irrigation). These factors affect
estimates of runoff, soil moisture, evapotranspiration, and sensible heat flux (land surface
temperatures) in irrigated areas. Therefore, it is important to be aware of the limitations and
combine with other products (e.g., NDVI or Actual Evapotranspiration (ETa) in irrigated areas)
when exploring water and energy balance. Even with improvements to meteorological forcing and
modeling parameterizations, errors will remain. Therefore, the 'convergence of evidence' approach
is beneficial and would be important when assessing hydro-meteorological hazards and associated
risks to food and water security. By making the data publicly available the broader food security and
water resources communities will be able to provide insights that can lead to improvements in our
understanding of the water and energy balance that can ultimately lead to improvements to food and
water security decision support systems.

**8. Author contribution**
JJ runs the code, updates websites, and archives routinely. DS maintains LISF code used in paper,
JJ, KA, DS, SP conducted model evaluation AM, KS, CPL, SK contributed to design of evaluation.
JR, MB, SP manage the data for USGS distribution. AH, JV provide feedback on data quality and
interpretation. AM prepared the manuscript with contributions from all co-authors.
**9. Acknowledgements**
The authors wish to acknowledge the original version of the Central Asia snow modeling with LIS6
performed at NOAA National Operational Hydrologic Remote Sensing Center by Greg Fall and
Logan Karsten.  USGS work was performed under U.S. Agency for International Development
(USAID), Bureau of Humanitarian Assistance (BHA) PAPA AID-FFP-T-17-00003 and USGS
contract 140G0119C0001. Any use of trade, firm, or product names is for descriptive purposes only
and does not imply endorsement by the U.S. Government. KS, AH, DS, JJ, NASA work was
performed under USAID BHA PAPA AID-FFP-T-17-00001. KS, AH acknowledge support from
the NASA Harvest Consortium (NASA Applied Sciences Grant No. 80NSSC17K0625). Computing
resources have been provided by NASA's Center for Climate Simulation (NCCS). Distribution of
data from the Goddard Earth Sciences Data and Information Services Center (GES DISC) is funded
by NASA's Science Mission Directorate (SMD). We thank NOAA CPC International Desk for use
of figures, and the NASA Land Information System Team for software support and development.
The authors also thank the USGS reviewer for comments that improved the quality of the
manuscript.

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
