# Peer review of "A Central Asia Hydrologic Monitoring Dataset for Food and Water Security Applications in Afghanistan"

_Earth System Science Data, 2021_

## Author Comment (AC1)

**RESPONSE TO REVIEWER #1**

*Author's response:* We really appreciate the comments from the reviewers and note some common themes: (1) *why were the parameters and forcing choices made*? R1.1, R1.2, R1.4, R2.6, R2.7 (2) *why were the evaluation methods chosen* R2.5, R2.8, R2.9. (3) some general questions that we think could be clarified by the motivation, and our interpretation of the scope of a data descriptor (4) *Addition of a clear 'limitations and future work' section* (5) Editorial comments: Figure legibility, figure placement, edits to intro and abstract.

Dear Reviewer #1, Thanks for the positive comments and we address your remarks and suggested revisions in the following.

1. It would be useful for the modeling community to understand a bit more about the data choices summarized in Table 1. Why were each of these parameter datasets selected? For example, the FAO soil texture dataset has been replaced by ISRIC in many applications, and the NCEP vegetation fraction dataset is a low resolution climatology. I don't ask that the authors change these settings, but given the status of FLDAS as an operational LDAS system with global scope I expect that the authors are in a good position to provide readers with some guidance regarding the choice of parameter sets. (Also: a minor note on Table 1—the row labels could be improved: why is the FAO soil data simply listed as "parameters" and the snow albedo simply listed as "albedo"? It looks like words were dropped.)

Thanks for this comment. We propose to add some variation of the following text:

"The parameters and specifications listed in Table 1 are largely default settings defined by the Noah LSM community (NCAR Research Applications Library, 2021). Ongoing research aims to identify where model output performance can be improved with parameter updates. Evaluating parameter updates had similar challenges as evaluating input forcing described in Section X: without reliable reference data it is difficult to determine a "best" input. For example, we have explored changing soil parameters from FAO to International Soil Reference and Information Centre (ISRIC) SoilGrids database (Hengle et al. 2017).  However this change did not result in improvements in streamflow statistics in Southern Africa, nor for soil moisture anomaly's ability to represent drought events. We expect similar results in Afghanistan where e.g., streamflow will be sensitive to a change in soil parameters and the challenge will be to obtain referenced data to evaluate if there is an improvement. Moreover, our model runs at 0.1 and 0.001 degrees may not fully exploit the added value of the 250m soil grids (noted in Ellenburg et al. (2021)) for a LIS application in East Africa.

Vegetation parameters are also potential sources of improvement whose importance to LDAS hydrologic estimates has been highlighted (e.g., Miller et al. 2006). We have found the NCEP estimates of green vegetation fraction (GVF) to be sufficient for this configuration of Noah 3.6, due to lack of demonstrated improvements with e.g. a time series of green vegetation fraction derived from NDVI which did not improve representation of droughts. However, Future FLDAS

Global and Central Asia, versions will be run with Noah-MP (Niu et al. 2007) which has vegetation options beyond that of Noah 3.6, and relies on Leaf Area Index rather than GVF, which will open up possibilities for choice of datasets to meet our application needs and potentially improve representation of the water balance.

*Re:* Also: a minor note on Table 1… Thank you, we have updated the rows in Table 1. Changing "Parameters" to "Soil Parameters", and combined average and max albelo into "Albedo"

> 2. Section 2.3: The authors explain how they used the different precipitation datasets, but I'm confused about the rationale. IMERG is not a particularly new product—it's been available for several years. Why does the system still use GDAS precipitation and only include IMERG as a data comparison? The IMERG Late (or Early) runs would have low enough latency for real-time monitoring applications, and the authors later note that they plan to integrate IMERG into the system. Is there a reason why this isn't already done? For example, some practical or product continuity advantage to using GDAS instead of IMERG?

Thanks for this question, and fits with themes we've identified in the reviews (1) why were the inputs chosen (2) better description of limitations and future work. We also propose to add text to the following sections:

*Section 1.2 Meteorological Background*
*… our choices of inputs must meet the following criteria: (a) provide a long historic record for contextualizing estimates in terms of departures from the mean (anomalies) (b) are low latency (< 1-month) for timely decision support (c) are familiar to the FEWS NET user-community and (d) prior evaluation by our team and the broader community.*
*In addition to these products (GDAS, CHIRPS, MERRA-2) the Integrated Multi-satellite Retrievals for the Global Precipitation Mission (IMERG), a NASA precipitation (Huffman et al., 2020) has emerged as a precipitation product that meets these requirements, since its period of record was extended back to 2000 as of version 6 which was released in 2019.*

*Section 3.4 Limitations and Future work:*
*As of 2019, with the release of IMERG v6, these data go back to 2000 as well. Prior to this change we began comparisons within our framework, described in (Kirschbaum et al. 2018) and found encouraging results. At that time however, the period of record was a limitation for computing anomalies. We now have an adequate period of record, and IMERG will be part of the upcoming FLDAS-Global and FLDAS-CA releases. We are also encouraged by the quality of IMERG: For example, Sarmiento et al. (2021) further explored the qualities of IMERG compared to CHIRPS over CONUS.*

*Other projects are also informing our understanding of how IMERG will perform within our system. One recent attempt to improve meteorological inputs in the region is from Ma et al. (2020) with the development of the AIMERG dataset that combines IMERG Final with APHRODITE (Asian Precipitation - Highly-Resolved Observational Data Integration Toward*

*Evaluation) rain-gauge derived product (Yatagai et al., 2012). Another promising development is CHIMES (Funk et al. 2022) derived from both CHIRPS and IMERG, whose developers have been exploring the strengths and limitations of these two datasets and how to fuse them to produce an optimal product.*

So, great point, we're certainly moving in that direction to incorporate the benefits of IMERG into the products co-authors are developing for the food/water security user community. We anticipate you may see these products on the USGS website in the next year, as well as associated technical documentation.

> 3. Figure 6: Can the authors comment on the fact that the coarse resolution global run appears to do better than the high resolution CA run in this data comparison? I find it surprising, given the presumed topographic sensitivity of SCA.

Thanks for this comment. The categorical statistics do indicate that Central Asia (GDAS) tends to have both a higher probability of detection AND false alarm rate, indicating higher mean than MODIS SCF and Global (CHIRPS). Figure 6 emphases the high mean and apparent better correspondence of the global run. This may be further exacerbated by the spatial averaging over the basin.

For an alternative perspective we conducted a pixel-wise NIC comparison of Central Asia, Global run's estimates of evapotranspiration vs SSEBop ET. We found that in general, GDAS derived estimates of ET consistently performed better over Afghanistan in terms of pixel-wise correlation with SSEBop ET (Figures below). The modeled estimates of surface soil moisture vs SMAP were more mixed depending on the month.

In general, given the lack of clarity on "best" product, our approach is to highlight the potential benefit of CHIRPS as highlighted in other literature (as well as its familiarity to the food security community), while GDAS has a benefit for our applications of being a low latency and higher spatial resolution for routine monitoring.

Given the potential for confusion, rather than adding clarify/information to our description we will *remove Figure 6.*

[Figure]

Figure R1. Normalized Information Contribution (NIC) comparison between Global (CHIRPS), and Central Asia (GDAS) monthly evapotranspiration estimates and SSEBop. Red indicates the reference dataset (SSEB) has relatively higher correlation with the Central Asia datastream and blue indicates relatively higher correlation of the Global runs.

Normalized Information Contribution (NIC) metric (following Sarmiento et al 2021) as used to compare the differences between the simulations using the Simplified Surface Energy Balance for operations (SSEBop) (Senay et al. 2013). The NIC used the simulated monthly anomalies for every month and correlated those with the SSEB monthly anomalies.

1. The authors note various limitations and potential areas of improvement throughout the paper. I would find it useful for this information to be included in a short section near the end of the paper on "Limitations and Future Work" that could describe ongoing FLDAS development activities. While the future work isn't a necessary component of this data description paper, it would be valuable for the reader to have this information when considering adopting FLDAS to support research or operations.

Good suggestion. We have changed Section 3.4 Limitations and Future Developments to include more discussion beyond Table 3 'pros and cons'

We now include discussion of the IMERG dataset, and future plans for continued evaluation and incorporation into future FLDAS versions (see response #2)
As previously mentioned regarding parameters, FLDAS Central Asia and Global will be transitioning to Noah-MP. This will allow for improved representation of glaciers and groundwater. This will also necessitate the use of different parameters e.g., LAI, as well as the potential to explore different parameter sets like ISRIC soils.

We note that our particular configuration has been developed within a specific set of constraints and for a specific user community that gives weight to length of record, latency, and familiarity with or compatibility with other routinely used products. We encourage the data user to consider how these factors may influence their particular use of and interpretation of the dataset. We also welcome feedback regarding the performance of these data when confronted with other questions, regions and reference datasets.

---

## Author Comment (AC2)

**REVIEWER #2**

*Author's response:* We really appreciate the comments from the reviewers and note some common themes: (1) *why were the parameters and forcing choices made*? R1.1, R1.2, R1.4, R2.6, R2.7 (2) *why were the evaluation methods chosen* R2.5, R2.8, R2.9. (3) some general questions that we think could be clarified by the motivation, and our interpretation of the scope of a data descriptor (4) *Addition of a clear 'limitations and future work' section* (5) Editorial comments: Figure legibility, figure placement, edits to intro and abstract.

1. Title: The title is misleading since the majority (if not all) the content focuses on Afghanistan. I would've been okay if the title was "A Hydrologic Monitoring Dataset for Food and Water Security Applications in Afghanistan" instead. I do appreciate the fact the system is setup for both globally and for the Central Asia domain, but there are no tests to corroborate its performance outside Afghanistan presented in the manuscript.

Thanks for this comment. We propose a new title that reflects that the data is available for all Central Asia, but our motivation/application is Afghanistan:

*A Central Asia Hydrologic Monitoring Dataset for Food and Water Security Applications in Afghanistan*

2. Abstract: The abstract is written quite general with results being presented rather vaguely

Thanks for this comment, we re-wrote the abstract to better reflect the criterion put forth by the journal for a data descriptor (significance, uniqueness of these data, usefulness for future interpretation, and completeness) as well as updates during the review process (additional content and re-organization of introduction, and framing/motivation).

From the Hindu Kush Mountains to the Registan desert, Afghanistan is a diverse landscape where droughts, floods, conflict, and economic market accessibility pose challenges for agricultural livelihoods and food security. The ability to remotely monitor environmental conditions is critical to support decision making for humanitarian assistance. The FEWS NET Land Data Assimilation System (FLDAS) global and Central Asia data streams provide information on hydrologic states for routine integrated food security analysis. While developed for a specific project these data are publicly available and useful for other applications requiring hydrologic estimates of snow water equivalent, soil moisture, runoff and other variables representing the water and energy balance. The unique aspects of these data are their suitability for routine monitoring and a historic record for computing relative indicators of water availability. Specifically, the global stream is available at ~1 month latency, monthly average outputs on a 10 km$^2$ grid from 1982-present. The second data stream, Central Asia, at ~1 day latency, provides daily average outputs on a 1 km$^2$ grid from 2000-present. This paper describes the configuration of the two FLDAS data streams, background on the software modeling framework, selected meteorological inputs and parameters, results from previous evaluation studies as well as a summary of strengths and limitations for future users. We

provide additional analysis of precipitation and snow cover over Afghanistan, and an example of how these data are used in integrated food security analysis. These data are hosted by NASA and US Geological Survey data portals for use in new and innovative studies and applications that may improve understanding of this important region.

3. Introduction: It is rather unusual to begin a section with the figures without any context

Thanks for this comment. We have re-organized the introduction so that the sections begin with text as follows.

1.0 Introduction:
From the Hindu Kush Mountains to the Registan desert, Afghanistan is a diverse landscape where droughts, floods, conflict, and economic market accessibility pose challenges for agricultural livelihoods and food security. The ability to remotely monitor environmental conditions and develop "Climate Services" is critical to support decision making for economic development and humanitarian assistance….

1.1 Afghanistan Weather and Climate
Central Asia, a region that includes Afghanistan, is water-scarce receiving roughly 75% of its annual precipitation during November–April … Figures 1a [precip] & Figure 1b [temperature]

4. Introduction: The section lacks a proper introduction within a broader context and motivation, both in terms of the region and in terms of efforts to predict land surface variables with modeling and remote sensing products

Thanks for this comment, and the opportunity to better motivate the work for the readers and connect to the theme of Climate Services and provide a more in-depth introduction to efforts that have previously conducted evaluation on inputs and outputs of land surface variables & remote sensing in the region. We think this better reflects our motivation, which is the development of a dataset for monitoring, rather than a focus on resolving uncertainties in the water balance. Here is new version of the introduction with new text highlighted in blue.

**1 Introduction**

From the Hindu Kush Mountains to the Registan desert, Afghanistan is a diverse landscape where droughts, floods, conflict, and economic market accessibility pose challenges for agricultural livelihoods and food security. The ability to remotely monitor environmental conditions and develop "Climate Services" is critical to support decision making for economic development and humanitarian assistance. A Climate Service, as defined by the World Meteorological Organization (WMO), and the Global Framework for Climate Services (Hewitt et al. 2012), is a "decision aid derived from climate information that assists individuals and organizations to improve decision making." Estimates of hydrologic variables relevant are water resources, agriculture, natural disaster risk reduction and more.

When hydrologic datasets are updated routinely the influences of climate variability and climate change can be incorporated into analysis by intermediary users[1]. These intermediate users (i.e., Climate Service providers) in turn can produce assessments of current and future needs for humanitarian assistance or other applications. Several case study examples can be found in McNally et al. 2019 that describe the co-production of Climate Services from hydrologic and agricultural Earth Observations. One such example, relevant to this data descriptor, is the Famine Early Warning System Network (FEWS NET), whose food security analysts combine environmental information, largely from remote sensing and earth system models, with nutrition, livelihoods and markets and trade to provide decision support to US Agency for International Development (USAID) Bureau of Humanitarian Assistance. Further discussion of the co-production of Climate Services can be found in the literature e.g. Vincent et al. 2018, and FEWS NET Climate Services (blog post).

This paper describes the FEWS NET Land Data Assimilation System (FLDAS) hydrologic modeling system's global and Central Asia data streams, which are designed for food and water security applications. Specifically, the inputs (e.g., precipitation) and resulting data streams (e.g., snow water equivalent) (a) provide a long historic record for contextualizing estimates in terms of departures from the mean (anomalies) (b) are low latency (< 1-month) for timely decision support (c) are familiar to the FEWS NET user-community. While these data are tailored to specific needs, this paper describes the data streams to enable their use by a broader community of researchers or Climate Service practitioners.

The purpose of this data descriptor is four-fold: (1) describe the development of the moderate resolution, low latency FLDAS system to inform hydrologic monitoring for Central Asia, specifically Afghanistan, (2) increase awareness of these data resources which are intended to be a public good, (3) demonstrate how our methods inform critical investigations that ultimately improve general understanding of water resources in this important region of the world, and (4) advocate for a convergence of evidence approach to hydrologic monitoring in locations where all sources of information contain some level of uncertainty.

An outline of this data descriptor is as follows. First, we'll provide Background on Afghanistan Weather and Climate. Then review previous studies that have conducted evaluations of the meteorological inputs and hydrologic outputs of Land Data Assimilation Systems in the Central Asia and High Mountain Asia region. In section 2 (Methods) we describe the hydrologic modeling system, parameters and meteorological inputs and model outputs. In the Results (section 3) we report comparisons to other precipitation estimates, as well as comparisons of modeled snow estimates to remotely sensed snow observations and find generally good agreement. Finally, we describe an application (section 4) of these data to the Afghanistan drought of 2018.
* * *
[1] the WMO defines intermediate (intermediary) users as those who transform climate information into a climate service

**1.1 Afghanistan Weather and Climate**

Central Asia, a region that includes Afghanistan, is water-scarce receiving roughly 75% of its annual precipitation during November–April (Oki and Kanae, S., 2006). In Afghanistan, rainfall is highest in the northeast Hindu Kush Mountains and decreases toward the arid southwest Registan Desert (Fig. 1a). Temperature follows a similar pattern with cooler temperatures in the high elevation and wetter northeast and warmer temperature in the south, and southwest (Fig. 1b). Regional precipitation is strongly influenced by the El Niño-Southern Oscillation (ENSO). La Niña condition are associated with below average precipitation (FEWS NET, 2020b) and El Niño conditions associated with above average precipitation (Barlow et al., 2016; Hoell et al., 2017; Rana et al., 2018; Hoell et al., 2018, 2020; FEWS NET, 2020a). Other dynamical factors with an important influence on precipitation include orography, storm tracks, and the Madden–Julian oscillation (MJO) (Barlow et al., 2005; Nazemosadat and Ghaedamini, 2010; Hoell et al., 2018). The last several years have experienced a number of ENSO events, with recent La Niña events in 2016-17, 2017-18, and 2020-2021 (NOAA CPC ENSO Cold & Warm Episodes by Season, 2021) that corresponded to droughts (FEWS NET, 2017b, 2018c, 2021).

[Figure]

Figure 1a. Mean annual precipitation in Afghanistan from 1991-2020, overlayed on province boundaries. Map (USGS Knowelge Base, 2021) with data from Funk et al. (2015).

[Figure]

**Afghanistan Historical Average Maximum Temperature**

Jan - Dec, 1986 - 2015

0    90    180        360 Kilometers

Map Produced by: USGS/EROS
Data Source: Climate Hazards InfraRed (maximum)
Temperature with Stations (CHIRTSmax) (USGS/UCSB)

**Temperature (°C)**

| | |
|---|---|
| < 8 | 16 - 18 | 26 - 28 |
| 8 - 10 | 18 - 20 | 28 - 30 |
| 10 - 12 | 20 - 22 | 30 - 32 |
| 12 - 14 | 22 - 24 | > 32 |
| 14 - 16 | 24 - 26 | water |

Figure 1b. Average maximum monthly temperature from (1986-2015), overlayed on province boundaries. Map (USGS Knowelge Base, 2021) with data from Verdin et al. (2020).

Despite Afghanistan's semi-arid climate, agriculture is an important sector, contributing 23% of the gross domestic product and employing 44% of the national labor force (CIA World Factbook). High mountain snowpack and snowmelt runoff are important for agricultural water supply, and according to the Famine Early Warning Systems Network (FEWS NET, 2018b) is responsible for "providing over 80% of irrigation water used. The timing and duration of the snowmelt is a key factor in determining the volume of irrigation water and the length of time that it is available, as well as its availability for use in marginal areas that experience [variable] rainfall." Therefore, routine hydrologic monitoring, with a particular emphasis on snow, is critical for tracking agricultural conditions and provides early warning for food insecurity.

**1.2 Hydrologic Data Availability and Uncertainty**

Remote sensing and models are important inputs to Climate Services. The challenge in the Central Asia and Afghanistan region however is that there is considerable uncertainty in estimates of meteorological inputs, model parameters and model estimates given the lack of in-situ environmental observations. One project that has explored this extensively is the NASA

High Mountain Asia project (https://www.himat.org/) that asked, "What is driving changes in hydrology and cryosphere in the High Asia region?" and guided by sub-teams focusing on model validation and data assimilation, cryosphere dynamics and water budget processes. We will provide a summary of literature from this project and others guided the configuration and interpretation of the FLDAS Central Asia and Global runs.

A primary challenge to producing and evaluating hydrologic estimate is sparse in-situ precipitation observations that lead to uncertainty in gridded and satellite-based precipitation estimates. Precipitation station observations are used for (a) bias correction of satellite estimates and (b) validation of gridded products. In terms of gridded dataset development, Hoell et al. (2015) describe lack of station observations in Afghanistan, Iraq and Pakistan and how complex topography in these locations makes this issue particularly problematic. Barlow et al. (2016) also highlight the station availability across the region and how that influences uncertainties in the Global Precipitation Climatology Center (GPCC) version 6 dataset over Central Asia (Fig. 2a) and specifically Afghanistan over time (Fig. 2b).

[Figure]

[Figure]

Figure 2. a) Station data availability underlying the GPCC version 6 dataset, for the 1951–2010 period, on the 0.5°-resolution grid over Central Asia. b) Number of Stations used as input to GPCC rainfall dataset in Afghanistan.

One approach for remote sensing and model evaluation, given the lack of in-situ observations is to compare multiple input datasets, especially precipitation, and evaluate the water balance as a whole to take advantage of independent observations from the different components (e.g. evapotranspiration, soil moisture, streamflow). Particularly relevant to this work are Yoon et al. (2019) and Ghatak et al. (2018) which we refer readers and data users to to appreciate the uncertainties in inputs, outputs and derived products and climate services over Afghanistan and the broader Central Asia region.

With respect to precipitation evaluation Ghatak et al. (2018) compare four unique precipitation data sources: daily Climate Hazards Infrared Precipitation with Stations (CHIRPS) product (Funk et al. 2015), NOAA's Global Data Assimilation System (GDAS) (Derber et al., 1991), and two estimates from NASA's Modern Era Reanalysis for Research and Applications version 2 (MERRA-2) (Gelaro et al., 2017).   These products were compared to APHRODITE (Asian Precipitation - Highly-Resolved Observational Data Integration Toward Evaluation) rain-gauge derived product (Yatagai et al., 2012). They find that Annual CHIRPS and GDAS precipitation estimates performed similarly over [Afghanistan] with respect to APHRODITE in terms of bias and root mean squared error (RMSE). CHIRPS had a higher correlation with APHRODITE. Ghatak et al. (2018) further evaluated the quality of rainfall inputs based on the performance of evapotranspiration and other derived outputs. The authors caution that "available gridded precipitation estimates based on in situ data may systematically underestimate precipitation in mountainous regions and that performance of gridded satellite-derived or modeled precipitation estimates varies systematically across the region."

Yoon et al. (2019) compare precipitation estimates from ten different products including APHRODITE, CHIRPS, GDAS, and MERRA2, across a broad region, covering a small portion of Afghanistan. They find that all datasets generally capture the spatial pattern rainfall and that products tend to agree more at high elevations, where it is unlikely there are station

observations. More specifically, they found GDAS to have a higher mean precipitation than CHIRPS, which was not surprising given concerns that station corrected datasets inherit a low bias from sparse gauge data. From the original ten precipitation products Yoon et al. went on to compare outputs from land surface models driven by CHIRPS, MERRA2, GDAS and ECMWF.

To summarize, GDAS, CHIRPS and MERRA-2 were chosen for our system based on our project requirements of (a) a sufficiently long historic record for contextualizing estimates in terms of departures from the mean (anomalies) (b) low latency (< 1-month) for timely decision support (c) familiar to the FEWS NET user-community. As well as prior evaluation by our team and the broader community. In addition to these products the Integrated Multi-satellite Retrievals for the Global Precipitation Mission (IMERG), a NASA precipitation (Huffman et al., 2020) has emerged as a precipitation product that meets these requirements, since its period of record was extended back to 2000 as of version 6 which was released in 2019. We will a describe IMERG, GDAS, and MERRA-2 comparison in the Results (Section 3).

The known uncertainties in precipitation datasets are the rationale behind the requirement for "(a) a sufficiently long historic record for contextualizing estimates in terms of departures from the mean (anomalies)". Schiemann et al. (2008) find that gridded precipitation estimates can qualitatively identify large scale spatial distribution of precipitation, seasonal cycle and interannual variability (i.e., wet and dry years) across Central Asia. Long term estimates of rainfall from satellite derived products, as well as derived historic time series from hydrologic modeling, can be used as a baseline of "observations," from which we can have a sense of relative conditions, i.e., anomalies and variability. When this historical record is harmonized with a routine monitoring system, current conditions can be placed in historical context. Anomaly-based representation of hydrologic extremes can provide confidence in modeled estimates that have the potential to influence agricultural, water resources and food security outcomes.

In addition to precipitation other meteorological inputs are important for accurate hydrologic estimates. Yoon et al. (2019) conduct an intercomparison of near surface air temperature (Tair) estimates from three model analysis products (ECMWF, GDAS, and MERRA2). They noted an upward trend in GDAS temperature, as well as consistently higher temperature in MERRA2.

From a Climate Services perspective, the reliance on the representation of relatively wet and dry conditions, as well as a "convergence of evidence" provide useable information despite the above-mentioned uncertainties. A convergence of evidence approach that draws on (quasi-)independent sources of information is useful to understand actual conditions. For convergence of Earth observations, hydrologic models can generate ensembles of historic, current or future estimates of snow, streamflow, soil moisture, and evapotranspiration which can then be compared to satellite derived estimates of surface water (e.g. McNally et al., 2019), soil moisture (e.g. McNally et al., 2016), vegetation conditions and evapotranspiration (e.g. Pervez et al., 2021; Jung et al., 2019), snow cover (e.g. Arsenault et al., 2014), in situ stream flow (e.g. Jung et al., 2017) and others. Hydrologic estimates can also be compared to outcomes in crop

production e.g. (McNally et al., 2015; Davenport et al., 2019; Shukla et al., 2020), and nutrition, health, and food security (e.g. Grace and Davenport, 2021) to provide a qualitative understanding of both hydrologic model performance and conditions on the ground. In this paper we provide an example of 2018 where drought conditions were associated with crisis levels of acute food insecurity over most of Afghanistan (FEWS NET, 2018c).

5. Section 2.2: Precipitation is mentioned as the most important input. However, I found the authors could have done a better job comparing multiple products (e.g., ERA-Land, MSWEP, and others). The comparison seems rather limited. It also gives the impression that precipitation is the only meaningful forcing to compare against other products. I'd assume temperature and radiation would play a role as well, especially if the focus is on getting snow water equivalent predictions. Why haven't the authors compared other forcing variables? How do we know they perform well in Afghanistan?

Thanks for this comment. Regarding the limited comparisons of other products we have now added additional background in Section 1.2 summarizing previous evaluation studies.

"Particularly relevant to this work are Yoon et al. (2019) and Ghatak et al. (2018) to which we refer readers and data users to appreciate the uncertainties in inputs, outputs and derived products and climate services over Afghanistan and the broader Central Asia region."

Meteorological forcing is known to be the primary source of uncertainty in Land Surface Model simulations (Kato et al. 2007). Thus, its evaluation is important to understand the quality of model outputs. For this reason, Ghatak et al. (2018) focus on precipitation analysis and compare four unique precipitation data sources: daily Climate Hazards Infrared Precipitation with Stations (CHIRPS) product (Funk et al., 2015), NOAA's Global Data Assimilation System (GDAS) (Derber et al., 1991), and two estimates from NASA's Modern Era Reanalysis for Research and Applications version 2 (MERRA-2) (Gelaro et al., 2017). These products were compared to APHRODITE (Asian Precipitation - Highly-Resolved Observational Data Integration Toward Evaluation) rain-gauge derived product (Yatagai et al., 2012). They find that Annual CHIRPS and GDAS precipitation estimates performed similarly over [Afghanistan] with respect to APHRODITE in terms of bias and root mean squared error (RMSE). CHIRPS had a higher correlation with APHRODITE. Ghatak et al. (2018) further evaluated the quality of rainfall inputs based on the performance of evapotranspiration and other derived outputs. The authors caution that "available gridded precipitation estimates based on in situ data may systematically underestimate precipitation in mountainous regions and that performance of gridded satellite-derived or modeled precipitation estimates varies systematically across the region."

Yoon et al. (2019) compare precipitation estimates from ten different products including APHRODITE, CHIRPS, GDAS, and MERRA2, across a broad region, covering a small portion of Afghanistan. They find that all datasets generally capture the spatial pattern rainfall and that products tend to agree more at high elevations, where it is unlikely there are station observations. More specifically, they found GDAS to have a higher mean precipitation than CHIRPS, which was not surprising given concerns that station corrected datasets inherit a low bias from sparse gauge data. [In the absence of a reference dataset to represent truth, Yoon et al. conducted an extended triple collocation analysis to generate estimates of RMSE, where CHIRPS and

APHRODITE had the lowest RMSE]. From the original ten precipitation products Yoon et al. went on to compare outputs from land surface models driven by CHIRPS, MERRA2, GDAS and ECMWF.

It also gives the impression that precipitation is the only meaningful forcing to compare against other products. I'd assume temperature and radiation would play a role as well, especially if the focus is on getting snow water equivalent predictions. Why haven't the authors compared other forcing variables? How do we know they perform well in Afghanistan?

Meteorological forcing is known to be the primary source of uncertainty in Land Surface Model simulations (Kato et al. 2007). Thus, its evaluation is important to understand the quality of model outputs. From the background that we've now provided we hope to communicate that the uncertainties in the forcing, to a certain extent, preclude being able to assess how sensitive the useful outputs are (e.g. SWE anomalies) to the differences in temperature, radiation, wind inputs. As research progresses these analyses will be more relevant and important for diagnosing errors.

*We have added more background regarding temperature.* Yoon et al. conduct an intercomparison of near surface air temperature (Tair) estimates from three model analysis products (ECMWF, GDAS, and MERRA2). They noted an upward trend in GDAS temperature, as well as consistently higher temperature in MERRA2.

We also conducted temperature analysis comparing GDAS and MERRA2, specifically over the Afghanistan domain. We confirmed the upward trend in GDAS precipitation, where MERRA-2 is consistently warmer before 2010 and find that GDAS and MERRA-2 temperature estimates are of similar magnitude 2011-2020.

[Figure]

Figure R2. Afghanistan spatially average Air temperature estimates (2000-2020) from Central Asia (GDAS) and Global (MERRA2) datastreams.

The upward trend in temperature (as mentioned in Yoon et al) from 2000-2020 is more evident in all seasons except for January-March. This non-stationarity is attributed to primarily changes in the spatial resolution of this NOAA operational dataset over time, and secondarily to other changes in the analysis.

6. L194-195: How did the authors find GDAS and CHIRPS appropriate? Any preliminary tests they had carried out? Can the authors be more specific here?

Thanks for this question, we've added additional background for the readers regarding both our criteria for choosing the precipitation inputs as well as results from other evaluation studies.

*Section 1.2 Meteorological Background*
*With respect to criteria:*
… our choices of inputs must meet the following criteria: (a) provide a long historic record for contextualizing estimates in terms of departures from the mean (anomalies) (b) are low latency (< 1-month) for timely decision support (c) are familiar to the FEWS NET user-community and (d) prior evaluation by our team and the broader community.

*With respect to precipitation evaluation:*
Ghatak et al. 2018 compare four unique precipitation data sources: daily Climate Hazards Infrared Precipitation with Stations (CHIRPS) product (Funk et al., 2015), NOAA's Global Data Assimilation System (GDAS) (Derber et al., 1991), and two estimates from NASA's Modern Era Reanalysis for Research and Applications version 2 (MERRA-2) (Gelaro et al., 2017).   These products were compared to APHRODITE (Asian Precipitation - Highly-Resolved Observational Data Integration Toward Evaluation) rain-gauge derived product (Yatagai et al., 2012). They find that Annual CHIRPS and GDAS precipitation estimates performed similarly over [Afghanistan] with respect to APHRODITE in terms of bias and root mean squared error (RMSE). CHIRPS had a higher correlation with APHRODITE.

Yoon et al. (2019) compare precipitation estimates from ten different products including APHRODITE, CHIRPS, GDAS, and MERRA2, across a broad region, covering a small portion of Afghanistan. They find that all datasets generally capture the spatial pattern rainfall and that products tend to agree more at high elevations, where it is unlikely there are station observations. More specifically, they found GDAS to have a higher mean precipitation than CHIRPS, which was not surprising given concerns that station corrected datasets inherit a low bias from sparse gauge data.

From our own analysis, both products are well correlated >0.9 at the monthly and annual timesteps (Table 2). They qualitatively compare well with each other and MODIS Snow Covered Fraction (Figure 6). And both datasets meet our criteria for a sufficiently long historical record for computing anomalies, < 1 month latency, and familiarity with the FEWS NET community (CHIRPS precipitation and GDAS meteorological forcings are used in several other products). We hope that our additional emphasis on this criteria in the introduction and methods will appropriately frame our motivation for the reader.

7. L199-200: The authors indicate that daily CHIRPS data need to be converted to sub-daily. There are other global products which are already sub-daily. Have the authors considered using those to bypass any further temporal disaggregation steps which could further introduce errors?

Thanks for this comment. First, we clarify in the methods that the downscaling step is required because water and energy balances are calculated sub-daily.

In Background Section 1.2 we now describe in more detail Yoon et al. 2019 and Ghatak et al. 2018 comparisons with GDAS and MERRA2 precipitation that are sub-daily, which we hope communitates to the reader the relatively good performance of CHIRPS. In addition these studies also used CHIRPS, which explicitly or implicitly required temporal downscaling to drive Land Surface Models. These CHIRPS forced model runs were shown to perform well in terms of e.g. ET comparisons. Perhaps more to the reviewer's point.

We also augment discussion in Limitations and Future Work on IMERG, also a sub-daily product that will not require the additional temporal downscaling step.

"As of 2019, with the release of IMERG v6, these data go back to 2000 as well. Prior to this change we began comparisons within our framework, described in (Kirschbaum et al. 2018) and found encouraging results. At that time however, the period of record was a limitation for computing anomalies. We now have an adequate period of record, and IMERG will be part of the upcoming FLDAS-Global and FLDAS-CA releases. We are also encouraged by the quality of IMERG: For example, Sarmeiento et al. (2021) further explored the qualities of IMERG compared to CHIRPS over CONUS.

Other projects are also informing our understanding of how IMERG will perform within our system. One recent attempt to improve meteorological inputs in the region is from Ma et al. (2020) with the development of the AIMERG dataset that combines IMERG Final with APHRODITE (Asian Precipitation - Highly-Resolved Observational Data Integration Toward Evaluation) rain-gauge derived product (Yatagai et al., 2012). Another promising development is CHIMES (Funk et al. 2022) derived from both CHIRPS and IMERG, whose developers have been exploring the strengths and limitations of these two datasets and how to fuse them tto produce an optimal product.

8. Section 3.1: Perhaps I am naive with the FLDAS system but how does comparing gridded precipitation give an indication of performance of the system. My understanding (and I can be wrong here) is that FLDAS is an uncoupled system relative to the atmosphere, so precipitation is forcing/input variable rather than diagnostic or prognostic. Can the authors clarify why the comparison is needed and how they can link with the performance of their system?

Thanks for this comment. You're correct that the FLDAS is an uncoupled system. We understand that in locations where there is confidence (low uncertainty) in model inputs then one can focus on evaluating sensitivity and performance of the model outputs with respect to the model parameters and the parameterizations. e.g. you have 'true' rainfall but wish to evaluate your runoff generation parameterization.

In response to comment #5 We have provided additional information in the introduction/background. We now better explain to the reader that large uncertainties exist in all of the components of the water budget estimates, beginning with the precipitation.

The revised introduction better explains that our motivation is to produce a dataset that can be used as input to 'Climate Services' and has been guided by previous studies that have determined credible model configurations (inputs, parameters, model). We then demonstrate that these

model configurations are indeed credible given their routine use in Climate Services and decision support. We do provide additional information on the precipitation inputs in particular to communicate to the reader inherent challenges of producing useful hydrologic estimates in this region.

9. Figure 4 and Table 2: Linear correlation coefficient (R) at monthly and annual scales are expected to give relatively good performance and mainly tracks the seasonal and major year-to-year variability, respectively. Since the authors stressed the sub-daily aspect of the product, how does the system compare with other daily and sub-daily precipitation products over Afghanistan? In addition, there is no metric referring to magnitude of rainfall as R relates mainly with this coarse temporal dynamics. The authors should consider looking at some "residual" metric (MAE, RMSE, MSE, …)

We've revised how we frame the sub-daily aspect of the forcings (Line ~320).

We note this step in our methodology because water and energy balances are solved on a sub-daily timestep. However, for Central Asia we don't have sufficient reference data available to assess the importance of sub-daily precipitation distribution, as was demonstrated by Sarmiento et al. (2020) of the United States where adequate reference data is available.

We also provide additional information in the background/introduction regarding the known uncertainties in monthly and annual precipitation estimates. Lack of in-situ reference data limits the ability to perform evaluation on a sub-daily time step.

Regarding the magnitude of rainfall, we also now summarize results from Yoon et al. (2018) and Ghatak et al. (2019) who were able to conduct relative comparisons against e.g. gauge derived APHRODITE rainfall estimates. They caution however, that these data should not be interpreted as 'truth' and given the spatial distribution of gauges, and the apparent underestimation of ET and streamflow, that these 'reference' datasets likely have a low bias. Future work in the community will help move toward more quantitative evaluation statistics. And this paper describes an available dataset, with known limitations that guide its application (e.g. in the use of relative indices like Snow Water Equivalent anomalies, rather than absolute estimates of water availability).

10. Figures 5 and 6: Notice that up until this point, the reader has no idea about the location of these Afghan basins (no map is presented). In addition, there are not a single evaluation metric presented/discussed in this sub-section, the interpretation of the results seems to be only visual.

Thanks for this comment, we now include a map (Figure 3) in the results section that shows the location of basins

[Figure]

Figure 3. Map of major river basins in Afghanistan used in the snow covered fraction analysis.

11. I found the example of application 2017-2018 wet season only for Afghanistan to be very limited when disseminating the global and Central Asia product as claimed by the authors. This example does not cover all aspects of a comprehensive evaluation and assessment of the performance of this system. How do we know the system works for normal years or anomalous wet periods? How about for other regions outside Afghanistan domain. I think it is very dangerous to extrapolate such limited results to larger domain and to other hydrometeorological conditions. I also found it strange the fact that impacts of drought on agriculture are mentioned by the authors but no analysis of soil moisture from FLDAS is provided directly to the readers. *The authors should present a much more thorough assessment in my opinion.*

We appreciate and agree with the reviewer's concern for the potential extrapolation of results. We have included explicit caution for users of the data regarding the challenges & uncertainties for data in this region in the Section 3.4 "limitations & future work" section before this 4.0 Applications section. We have also better described previous literature on evaluation and uncertainties associated with these data.

The intent of this section is to demonstrate the 'significance of the dataset' specifically the criteria that it is being used in a Climate Services/*decision support context for food and water*

*security applications*, rather than a comprehensive evaluation.  This presentation is now better framed in the introduction where we now describe these data are motivated by the need for "Climate Services" where relative estimates and routinely updated information can be applied to different questions.

How do we know the system works for normal years or anomalous wet periods? How about for other regions outside the Afghanistan domain?

We do have some anecdotal examples of it working in wet periods (e.g. Widespread snowfall in Afghanistan). The reader could also refer to the products in Table 4. To confirm performance in normal or anomalous wet periods, the FEWS NET Afghanistan Seasonal Monitors highlight the use of these data since 2018.

However, we understand that the reviewer is likely hoping for a more quantitative analysis! We find that the development of a metric that would account for performance of a derived indicator is beyond the scope of this data descriptor e.g. categorical statistics (POD and FAR) for below-normal, normal, and above-normal years. This would require an independent reference dataset. We and other authors have attempted comparison with remotely sensed data (e.g. soil moisture, evaporation, total water storage, microwave snow estimates) but each of these data sources has its own set of errors that needs to be accounted for in the interpretation.

We hope that with the improvements to the introduction in terms of our motivation to provide inputs to Climate Services, as well as a review of prior evaluations better frames this section as a demonstration of the significance of these datasets, specifically that they are being applied in routine decision support.

12. Figure 10: Notice some of the text in the figure is too small to read.

We will separate out the figures so that the text is legible in the resubmission.

[Figure]

Figure 10a. Basins and provinces highlighted in the 2017-18 drought example.

[Figure]

Figure 10b. CHIRPS cumulative rainfall for 2017-18 vs average conditions for Daykundi Province. Figure from USGS EWX.

[Figure]

Figure 10c. CHIRPS cumulative rainfall for 2017-18 vs average conditions for Maydan Wardak Province. Figure from USGS EWX.

[Figure]

Figure 10d. Helmand Basin snow water equivalent (SWE) from the Central Asia data stream. The grey shading indicates the range of the minimum and maximum values, dashed blue line is the average, and black line is 2017-18. Figure from USGS EWX.

---

## Author Response (AR1)

*Author's response:* We really appreciate the comments from the reviewers and note some common themes: (1) *why were the parameters and forcing choices made*? R1.1, R1.2, R1.4, R2.6, R2.7 (2) *why were the evaluation methods chosen* R2.5, R2.8, R2.9. (3) some general questions that we think could be clarified by the motivation, and our interpretation of the scope of a data descriptor (4) *Addition of a clear 'limitations and future work' section* (5) Editorial comments: Figure legibility, figure placement, edits to intro and abstract. Thank you for taking the time to provide thoughtful feedback that has greatly improved our manuscript.

**RESPONSE TO REVIEWER #1**

Dear Reviewer, #1, Thanks for the comments and we address your remarks and suggested revisions in the following.

1. It would be useful for the modeling community to understand a bit more about the data choices summarized in Table 1. Why were each of these parameter datasets selected? For example, the FAO soil texture dataset has been replaced by ISRIC in many applications, and the NCEP vegetation fraction dataset is a low-resolution climatology. I don't ask that the authors change these settings but given the status of FLDAS as an operational LDAS system with global scope I expect that the authors are in a good position to provide readers with some guidance regarding the choice of parameter sets. (Also: a minor note on Table 1—the row labels could be improved: why is the FAO soil data simply listed as "parameters" and the snow albedo simply listed as "albedo"? It looks like words were dropped.)

Thanks for this comment. Added the following text in Line 272-292:

The parameters and specifications listed in Table 1 are largely default settings defined by the Noah LSM community (NCAR Research Applications Library, 2021). Ongoing research aims to identify where model output performance can be improved with parameter updates. Evaluating parameter updates had similar challenges as evaluating input forcing described in Section 1.2: without reliable reference data it is difficult to determine a "best" input. For example, we have explored changing soil parameters from FAO to International Soil Reference and Information Centre (ISRIC) SoilGrids database (Hengl et al., 2017). This change did not result in improvements in streamflow statistics in southern Africa, nor in soil moisture anomalies' ability to represent drought events. We expect similar results in Afghanistan where, e.g., streamflow will be sensitive to a change in soil parameters and the lack of referenced data to evaluate if there is an improvement. Moreover, our model runs at 0.1 and 0.01 degrees may not fully exploit the added value of the 250m soil grids as noted in Ellenburg et al. (2021) for a LISF application in East Africa.

Vegetation parameters are also potential sources of improvement whose importance to LDAS hydrologic estimates has been highlighted (e.g., Miller et al., 2006). We have found the NCEP estimates of green vegetation fraction (GVF) to be sufficient for this configuration of Noah 3.6. We found that a time series of GVF derived from the Normalized Difference Vegetation Index

(NDVI) did not improve representation of droughts in eastern Africa. However, future FLDAS global and Central Asia versions can be run with Noah-Multi parameterization (Noah-MP) (Niu et al., 2011) which has multiple vegetation options and relies on either Leaf Area Index rather or GVF. This model update is expected to open possibilities for choice of datasets to meet our application needs and potentially improve representation of the water balance.

*Re:* Also: a minor note on Table 1... Thank you, we have updated the rows in Table 1. Changing "Parameters" to "Soil Parameters", and combined average and max albedo into "Albedo"

> 2. Section 2.3: The authors explain how they used the different precipitation datasets, but I'm confused about the rationale. IMERG is not a particularly new product—it's been available for several years. Why does the system still use GDAS precipitation and only include IMERG as a data comparison? The IMERG Late (or Early) runs would have low enough latency for real-time monitoring applications, and the authors later note that they plan to integrate IMERG into the system. Is there a reason why this isn't already done? For example, some practical or product continuity advantage to using GDAS instead of IMERG?

Thanks for this question and it fits with themes we've identified in the reviews (1) why the inputs were chosen (2) better description of limitations and future work. We also propose to add text to the following sections:

*So, great point, we're certainly moving in that direction to incorporate the benefits of IMERG into the products co-authors are developing for the food/water security user community. We anticipate you may see these products on the USGS website in the next year, as well as associated technical documentation.*

**1.2 Hydrologic Data Availability and Uncertainty** *Line 213-222*

To summarize, our experience and the literature have characterized uncertainties in available meteorological forcing for the region. GDAS, CHIRPS, and MERRA-2 were chosen for the FLDAS system based on our project requirements of (a) a sufficiently long historical record for contextualizing estimates in terms of anomalies (b) low latency (< 1-month) for timely decision support, (c) familiar to the FEWS NET user-community, and (d) prior evaluation by our team and the broader community. We note here and describe in more detail later that the Integrated Multi-satellite Retrievals for the Global Precipitation Mission (IMERG), a NASA precipitation product (Huffman et al., 2020) also meets these requirements, since version 6 which was released in 2019 (after these studies and initial FLDAS configuration). We will a describe IMERG, GDAS, and MERRA-2 comparison in the Results (Section 3).

*3.4 Limitations and Future Developments Line 474-493*
IMERG version 6 was released in 2019 and includes rainfall estimates processed back to 2000. Prior to this change we had found encouraging results when comparing the onset of rainy season using both IMERG Late Run and CHIRPS (Kirschbaum et al., 2016). However, at that time the period of record was a limitation for computing anomalies. We now have an adequate

period of record, and IMERG Late Run is planned to be part of the upcoming FLDAS global and FLDAS Central Asia releases. We are also encouraged by the quality of IMERG at the daily timestep when compared to CHIRPS over the United States where accurate reference data are available (Sarmiento et al., 2021).

In addition to IMERG other promising rainfall datasets are in development. Ma et al. (2020) have developed the AIMERG dataset that combines IMERG Final Run with the APHRODITE rain-gauge derived product (Yatagai et al., 2012).  Another promising dataset is CHIMES (Funk et al., 2022), a blend of CHIRPS and IMERG, whose developers have been exploring the strengths and limitations of these two datasets and their fusion to produce an optimal product.

With respect to other FLDAS developments, FLDAS global and Central Asia are planned to be transition to Noah-MP. This will allow for improved representation of snowpack and groundwater. This will also necessitate the use of different parameters, e.g., leaf area index, as well as the potential to explore different parameter sets like ISRIC soils.  In the meantime, multi-forcing and multi-model ensembles, and convergence of evidence with other remotely sensed data and field reports, are a viable approach for providing hydrologic estimates for various applications.

3. Figure 6: Can the authors comment on the fact that the coarse resolution global run appears to do better than the high-resolution CA run in this data comparison? I find it surprising, given the presumed topographic sensitivity of SCA.

Given the potential for confusion, rather than adding clarify/information to our description we *removed Figure 6.*

The categorical statistics do indicate that Central Asia (GDAS) tends to have both a higher probability of detection AND false alarm rate, indicating higher mean than MODIS SCF and Global (CHIRPS). Figure 6 emphases the high mean and apparent better correspondence of the global run. This may be further exacerbated by the spatial averaging over the basin.

We added some additional text regarding additional analysis we conducted comparing the Central Asia and global datastreams to other remotely sensed products:

Methods: Line 370-380

In addition to rainfall and snow comparisons, we conducted monthly pixel-wise comparison of Central Asia and the global run's estimates of evapotranspiration (ET) and soil moisture versus Operational Simplified Surface Energy Balance (SSEBop, (Senay et al., 2013)). ET and Soil Moisture Active Passive (SMAP) Level 3 (Entekhabi et al., 2010, 2016) using the Normalized Information Contribution (NIC) metric following Sarmiento et al., (2021). The analysis was performed for the period 2016-2021 to match the SMAP record. The NIC metric first computes anomaly correlations between the model runs and the reference dataset and then computes

the difference between the performance of each model run using a scale of -1 to +1 to highlight if the global or Central Asia data stream performs better with respect to the reference. To make the comparisons, the reference datasets (SMAP and SSEBop) were re-gridded to match the grid spacing and locations of the experiment model outputs.
Results: Line 436-443

In addition to precipitation and snow cover comparisons we conducted comparisons with remotely sensed soil moisture and ET (not shown). We found that in general, GDAS derived estimates of ET consistently performed better over Afghanistan in terms of pixel-wise anomaly correlation and NIC with SSEBop ET. Meanwhile, neither modeled estimate of soil moisture consistently outperformed the other with respect to SMAP. The ET results lend some support to the quality of the Central Asia data stream estimates. However, the lack of signal in the soil moisture comparisons suggests that more careful analysis of the model and remote sensing errors is required before drawing conclusions regarding which data stream is "best."

4. The authors note various limitations and potential areas of improvement throughout the paper. I would find it useful for this information to be included in a short section near the end of the paper on "Limitations and Future Work" that could describe ongoing FLDAS development activities. While the future work isn't a necessary component of this data description paper, it would be valuable for the reader to have this information when considering adopting FLDAS to support research or operations.

Good suggestion. We added *3.4 Limitations and Future Developments Line 474-493*

IMERG version 6 was released in 2019 and includes rainfall estimates processed back to 2000. Prior to this change we had found encouraging results when comparing the onset of rainy season using both IMERG Late Run and CHIRPS (Kirschbaum et al., 2016). However, at that time the period of record was a limitation for computing anomalies. We now have an adequate period of record, and IMERG Late Run is planned to be part of the upcoming FLDAS global and FLDAS Central Asia releases. We are also encouraged by the quality of IMERG at the daily timestep when compared to CHIRPS over the United States where accurate reference data are available (Sarmiento et al., 2021).

In addition to IMERG other promising rainfall datasets are in development. Ma et al. (2020) have developed the AIMERG dataset that combines IMERG Final Run with the APHRODITE rain-gauge derived product (Yatagai et al., 2012).  Another promising dataset is CHIMES (Funk et al., 2022), a blend of CHIRPS and IMERG, whose developers have been exploring the strengths and limitations of these two datasets and their fusion to produce an optimal product.

With respect to other FLDAS developments, FLDAS global and Central Asia are planned to be transition to Noah-MP. This will allow for improved representation of snowpack and groundwater. This will also necessitate the use of different parameters, e.g., leaf area index, as well as the potential to explore different parameter sets like ISRIC soils.  In the meantime,

multi-forcing and multi-model ensembles, and convergence of evidence with other remotely sensed data and field reports, are a viable approach for providing hydrologic estimates for various applications.

**RESPONSE TO REVIEWER #2**

1. Title: The title is misleading since the majority (if not all) the content focuses on Afghanistan. I would've been okay if the title was "A Hydrologic Monitoring Dataset for Food and Water Security Applications in Afghanistan" instead. I do appreciate the fact the system is setup for both globally and for the Central Asia domain, but there are no tests to corroborate its performance outside Afghanistan presented in the manuscript.

Thanks for this comment. We propose a new title that reflects that the data is available for all Central Asia, but our motivation/application is Afghanistan:

*A Central Asia Hydrologic Monitoring Dataset for Food and Water Security Applications in Afghanistan*

2. Abstract: The abstract is written quite general with results being presented rather vaguely

Thanks for this comment, we re-wrote the abstract to better reflect the criterion put forth by the journal for a data descriptor (significance, uniqueness of these data, usefulness for future interpretation, and completeness) as well as updates during the review process (additional content and re-organization of introduction, and framing/motivation).

From the Hindu Kush Mountains to the Registan desert, Afghanistan is a diverse landscape where droughts, floods, conflict, and economic market accessibility pose challenges for agricultural livelihoods and food security. The ability to remotely monitor environmental conditions is critical to support decision making for humanitarian assistance. The Famine Early Warning Systems Network (FEWS NET) Land Data Assimilation System (FLDAS) global and Central Asia data streams provide information on hydrologic states for routine integrated food security analysis. While developed for a specific project, these data are publicly available and useful for other applications that require hydrologic estimates of the water and energy balance. These two data streams are unique because of their suitability for routine monitoring, as well as a historical record for computing relative indicators of water availability. The global stream is available at ~1 month latency, monthly average outputs on a 10-km grid from 1982-present. The second data stream, Central Asia (30-100 °E, 21-56 °N), at ~1 day latency, provides daily average outputs on a 1-km grid from 2000-present. This paper describes the configuration of the two FLDAS data streams, background on the software modeling framework, selected meteorological inputs and parameters, and results from previous evaluation studies. We also provide additional analysis of precipitation and snow cover over Afghanistan. We conclude with an example of how these data are used in integrated food security analysis. These data are hosted by the National Aeronautics and Space Administration (NASA) and U.S. Geological

Survey data portals for use in new and innovative studies that will improve understanding of this region.

3. Introduction: It is rather unusual to begin a section with the figures without any context

Thanks for this comment. We have re-organized the introduction so that the sections begin with text as follows.

1.0 Introduction Lines 41-84:
From the Hindu Kush Mountains to the Registan desert, Afghanistan is a diverse landscape where droughts, floods, conflict, and economic market accessibility pose challenges for agricultural livelihoods and food security. The ability to remotely monitor environmental conditions is critical to support decision making for economic development, humanitarian assistance, water resource management, agriculture and more. Environmental datasets can be combined with socio-economic variables and transformed into customized products to support decision making. This is the definition of a 'climate service' (Hewitt et al., 2012).

1.1 Afghanistan Weather and Climate Lines 85-116
Central Asia, a region that includes Afghanistan, is water-scarce, receiving roughly 75% of its annual precipitation during November–April (Oki and Kanae, 2006). In Afghanistan, rainfall is highest in the northeast Hindu Kush Mountains and decreases toward the arid southwest Registan Desert **(Fig. 1a).** Temperature follows a similar pattern with cooler temperatures in the high elevation, wetter northeast and warmer temperatures in the south and southwest **(Fig. 1b).**

4. Introduction: The section lacks a proper introduction within a broader context and motivation, both in terms of the region and in terms of efforts to predict land surface variables with modeling and remote sensing products

Thanks for this comment. If you refer to the 'track changes' document, you'll see that we've re-written the entire intro and appreciate and the opportunity to better motivate the work for the readers. We connect to the theme of Climate Services and provide a more in-depth introduction to efforts that have previously conducted evaluation on inputs and outputs of land surface variables & remote sensing in the region. We think this better reflects our motivation, which is the development of a dataset for monitoring, rather than a focus on resolving uncertainties in the water balance. Below highlights the how we've reorganized to provide background on 3 distinct aspects of the data descriptor.

**1 Introduction**
From the Hindu Kush Mountains to the Registan desert, Afghanistan is a diverse landscape where droughts, floods, conflict, and economic market accessibility pose challenges for agricultural livelihoods and food security. The ability to remotely monitor environmental conditions is critical to support decision making for economic development, humanitarian assistance, water resource management, agriculture and more. Environmental datasets can be

combined with socio-economic variables and transformed into customized products to support decision making. This is the definition of a 'climate service' (Hewitt et al., 2012). **[for more see revised text…]**

**1.1 Afghanistan Weather and Climate**

Central Asia, a region that includes Afghanistan, is water-scarce, receiving roughly 75% of its annual precipitation during November–April (Oki and Kanae, 2006). In Afghanistan, rainfall is highest in the northeast Hindu Kush Mountains and decreases toward the arid southwest Registan Desert (Fig. 1a). Temperature follows a similar pattern with cooler temperatures in the high elevation, wetter northeast and warmer temperatures in the south and southwest (Fig. 1b). Regional precipitation is strongly influenced by the El Niño-Southern Oscillation (ENSO). La Niña conditions are associated with below average precipitation (FEWS NET, 2020b) and El Niño conditions are associated with above average precipitation (Barlow et al., 2016; Hoell et al., 2017; Rana et al., 2018; Hoell et al., 2018, 2020; FEWS NET, 2020a). Other factors with an important influence on precipitation include orography, storm tracks, and the Madden–Julian oscillation (Barlow et al., 2005; Nazemosadat and Ghaedamini, 2010; Hoell et al., 2018). The last several years have experienced several ENSO events, with recent La Niña events in 2016-17, 2017-18, and 2020-2022 (NOAA CPC ENSO Cold & Warm Episodes by Season, 2021) that corresponded to droughts (FEWS NET, 2017b, 2018c, 2021). **[for more see revised text…]**

**1.2 Hydrologic Data Availability and Uncertainty**

Remote sensing and models are important inputs to climate services (Qamer et al., 2019). In the Central Asia region, and especially Afghanistan estimates of meteorological inputs, and model parameters have considerable uncertainty due to sparse in situ environmental observations. To address these challenges, the NASA High Mountain Asia project (https://www.himat.org/) has broadly aimed to explore the driving changes in hydrology as well as model validation and data assimilation, and water budget processes from the Himalayas in the south and east to the Hindu Kush in the west. These efforts and other studies of satellite derived rainfall informed the configuration and interpretation of the FLDAS Central Asia and global data streams. **[for more see revised text…]**

5. Section 2.2: Precipitation is mentioned as the most important input. However, I found the authors could have done a better job comparing multiple products (e.g., ERA-Land, MSWEP, and others). The comparison seems rather limited. It also gives the impression that precipitation is the only meaningful forcing to compare against other products. I'd assume temperature and radiation would play a role as well, especially if the focus is on getting snow water equivalent predictions. Why haven't the authors compared other forcing variables? How do we know they perform well in Afghanistan?

*Thanks for this comment. Regarding the limited comparisons of other products, we have now added additional background in Section 1.2 summarizing previous evaluation studies. Line 143-183*

In the absence of abundant in situ observations, one approach for remote sensing and model evaluation is to compare multiple input datasets and evaluate the water balance. Independent observations from the different components of the water balance (e.g., evapotranspiration, soil moisture, streamflow) help constrain estimates. We provide some background here and refer readers and data users to literature from the NASA High Mountain Asia project, specifically Yoon et al. (2019) and Ghatak et al. (2018), who explored similar configurations to the FLDAS system. This background allows the reader to appreciate the uncertainties in inputs, outputs and derived products and climate services over Afghanistan and the broader Central Asia region.

Meteorological forcing is known to be the primary source of uncertainty in land surface model simulations (Kato and Rodell, 2007). Thus, its evaluation is important to understand the quality of model inputs and outputs. For this reason, Ghatak et al. (2018) compare four unique precipitation data sources: daily Climate Hazards center Infrared Precipitation with Stations (CHIRPS) (Funk et al., 2015), NOAA's Global Data Assimilation System (GDAS) (Derber et al., 1991), and two estimates from NASA's Modern Era Reanalysis for Research and Applications version 2 (MERRA-2) (Gelaro et al., 2017). They find that annual CHIRPS and GDAS precipitation estimates had similar bias and root mean squared error over Afghanistan with respect to APHRODITE (Asian Precipitation Highly Resolved Observational Data Integration Toward Evaluation) rain-gauge derived product (Yatagai et al., 2012). CHIRPS had a higher correlation with APHRODITE. Ghatak et al. (2018) further evaluated the quality of rainfall inputs based on the performance of evapotranspiration and other derived outputs. The authors caution that gridded precipitation estimates that have in situ inputs, like CHIRPS, may systematically underestimate precipitation in mountainous regions. We keep this consideration in mind when interpreting differences between FLDAS global and Central Asia data streams.

Yoon et al. (2019) compare precipitation estimates from 10 different products including APHRODITE, CHIRPS, GDAS, and MERRA-2, across a broad region of High Asia, including a portion of Afghanistan. They find that all datasets generally capture the spatial pattern of rainfall and that the products tend to agree more at high elevations, where it is unlikely there are station observations. Like Ghatak et al. (2018), they found CHIRPS and APHRODITE to have a lower average precipitation than GDAS, attributable to the incorporation of sparse gauge data.

It also gives the impression that precipitation is the only meaningful forcing to compare against other products. I'd assume temperature and radiation would play a role as well, especially if the focus is on getting snow water equivalent predictions. Why haven't the authors compared other forcing variables? How do we know they perform well in Afghanistan?

Lines 175-183: In addition to precipitation, other meteorological inputs are important for accurate hydrologic estimates. Yoon et al. (2019) conducted an intercomparison of near surface air temperature estimates from three model analysis products (European Centre for Medium-Range Weather Forecasts (ECMWF; Molteni et al., 1996), GDAS, and MERRA-2). They noted a statistically significant upward trends in GDAS and ECMWF temperature, as well as consistently higher temperatures in MERRA-2. We see the same pattern when averaging across Afghanistan.

Yoon et al. (2019) conclude that improvements in the meteorological boundary conditions would be needed to reduce the uncertainty in the terrestrial budget estimates. These sentiments are echoed in Qamer et al. (2019).

We also conducted temperature analysis comparing GDAS and MERRA2, specifically over the Afghanistan domain. And included the following the following text in Lines 320-328

The FLDAS models require additional meteorological inputs, including air temperature, humidity, radiation, and wind. The lower-latency Central Asia data stream uses GDAS 3-hourly meteorological inputs available from 2001-present at <1-day latency. For a longer historical record, the global data stream uses MERRA-2 (Gelaro et al., 2017) (1979-present) 1-hourly products with a two-week latency. Over the Afghanistan domain GDAS temperature has an upward trend, whereas MERRA-2 is consistently warmer before 2010. We find that GDAS and MERRA-2 temperature estimates are of similar magnitude during 2011-2020. Similar results were noted by Yoon et al. (2019) who found an upward trend in GDAS temperature, as well as consistently higher temperatures in MERRA-2 across a broad High Asia domain.

6. L194-195: How did the authors find GDAS and CHIRPS appropriate? Any preliminary tests they had carried out? Can the authors be more specific here?

Thanks for this question, we've added additional background for the readers regarding both our criteria for choosing the precipitation inputs as well as results from other evaluation studies.

*Section 1.2 Meteorological Background*
*With respect to criteria:*

*Lines 64-67:* The inputs (e.g., precipitation) and resulting hydrologic estimates (a) provide a 40+ year historical record for contextualizing estimates in terms of departures from average (i.e., anomalies), (b) are low latency (< 1-month) for timely decision support, and (c) are familiar to the food and water security user-community.

[revised manuscript text omitted]

7. L199-200: The authors indicate that daily CHIRPS data need to be converted to sub-daily. There are other global products which are already sub-daily. Have the authors considered using those to bypass any further temporal disaggregation steps which could further introduce errors?

Thanks for this comment. First, we clarify in the methods that the downscaling step is required because water and energy balances are calculated sub-daily. Lines 311-317

For this approach, we use a finer timescale MERRA-2 precipitation timescale as a reference dataset to represent an accurate diurnal cycle. We note that this step in our methodology facilitates the solving of FLDAS water and energy balances at a sub-daily timestep. However, for Central Asia we do not have sufficient reference data available to assess the importance of sub-daily precipitation distribution, as was demonstrated by Sarmiento et al. (2021) for the United States where adequate reference data are available.

In Background Section 1.2 we now describe in more detail Yoon et al. 2019 and Ghatak et al. 2018 comparisons with CHIRPS, GDAS and MERRA2 precipitation that are sub-daily, which we hope communicates to the reader the relatively good performance of CHIRPS, and justification for including it as a forcing. These studies also used CHIRPS, which explicitly or implicitly required temporal downscaling to drive Land Surface Models.

We also augment discussion in Limitations and Future Work on IMERG, also a sub-daily product that will not require the additional temporal downscaling step. Lines 474-480

IMERG version 6 was released in 2019 and includes rainfall estimates processed back to 2000. Prior to this change we had found encouraging results when comparing the onset of rainy season using both IMERG Late Run and CHIRPS (Kirschbaum et al., 2016). However, at that time the period of record was a limitation for computing anomalies. We now have an adequate period of record, and IMERG Late Run is planned to be part of the upcoming FLDAS global and FLDAS Central Asia releases. We are also encouraged by the quality of IMERG at the daily timestep when compared to CHIRPS over the United States where accurate reference data are available (Sarmiento et al., 2021).

8. Section 3.1: Perhaps I am naive with the FLDAS system but how does comparing gridded precipitation give an indication of performance of the system. My understanding (and I can be wrong here) is that FLDAS is an uncoupled system relative to the atmosphere, so precipitation is forcing/input variable rather than diagnostic or prognostic. Can the authors clarify why the comparison is needed and how they can link with the performance of their system?

Thanks for this comment. You're correct that the FLDAS is an uncoupled system. We understand that in locations where there is confidence (low uncertainty) in model inputs then one can focus on evaluating sensitivity and performance of the model outputs with respect to the model parameters and the parameterizations. e.g., you have 'true' rainfall but wish to evaluate your runoff generation parameterization.

In response to comment #5 We have provided additional information in the introduction/background. We now better explain to the reader that large uncertainties exist in all the components of the water budget estimates, beginning with the precipitation.

The revised introduction better explains that our motivation is to produce a dataset that can be used as input to 'Climate Services' and has been guided by previous studies that have determined credible model configurations (inputs, parameters, model). We then demonstrate that these model configurations are indeed credible given their routine use in Climate Services and decision support. We do provide additional information on the precipitation inputs to communicate to the reader inherent challenges of producing useful hydrologic estimates in this region.

9. Figure 4 and Table 2: Linear correlation coefficient (R) at monthly and annual scales are expected to give relatively good performance and mainly tracks the seasonal and major year-to-year variability, respectively. Since the authors stressed the sub-daily aspect of the product, how does the system compare with other daily and sub-daily precipitation products over Afghanistan? In addition, there is no metric referring to magnitude of rainfall as R relates mainly with this coarse temporal dynamics. The authors should consider looking at some "residual" metric (MAE, RMSE, MSE, …)

In response to this and comment #7 We've revised how we frame the sub-daily aspect of the forcings.

Lines 311-317: For this approach, we use a finer timescale MERRA-2 precipitation timescale as a reference dataset to represent an accurate diurnal cycle. We note that this step in our methodology facilitates the solving of FLDAS water and energy balances at a sub-daily timestep. However, for Central Asia we do not have sufficient reference data available to assess the importance of sub-daily precipitation distribution, as was demonstrated by Sarmiento et al. (2021) for the United States where adequate reference data are available.

Regarding the magnitude of rainfall, we also now summarize results from Yoon et al. (2018) and Ghatak et al. (2019) who were able to conduct relative comparisons against e.g. gauge derived APHRODITE rainfall estimates. They caution however, that these data should not be interpreted as 'truth' and given the spatial distribution of gauges, and the apparent underestimation of ET and streamflow, that these 'reference' datasets likely have a low bias. Given the lack of a strong baseline, particularly at sub-monthly timesteps, residual metrics may be misleading.

Future work in the community will help move toward more quantitative evaluation statistics. And this paper describes an available dataset, with known limitations that guide its application (e.g. in the use of relative indices like Snow Water Equivalent anomalies, rather than absolute estimates of water availability).

10. Figures 5 and 6: Notice that up until this point, the reader has no idea about the location of these Afghan basins (no map is presented). In addition, there are not a single evaluation metric presented/discussed in this sub-section, the interpretation of the results seems to be only visual.

Thanks for this comment, we now include a map (Figure 5) in the results section that shows the location of basins

[Figure]

Figure 5. Hydrologic basins used in the analysis of categorical statistics for snow covered fraction.

11. I found the example of application 2017-2018 wet season only for Afghanistan to be very limited when disseminating the global and Central Asia product as claimed by the authors. This example does not cover all aspects of a comprehensive evaluation and assessment of the performance of this system. How do we know the system works for normal years or anomalous wet periods? How about for other regions outside Afghanistan domain. I think it is very dangerous to extrapolate such limited results to larger domain and to other hydrometeorological conditions. I also found it strange the fact that impacts of drought on agriculture are mentioned by the authors but no analysis of soil moisture from FLDAS is

provided directly to the readers. The authors should present a much more thorough assessment in my opinion.

We appreciate and agree with the reviewer's concern for the potential extrapolation of results. We have included explicit caution for users of the data regarding the challenges & uncertainties for data in this region in the Section 3.4 "limitations & future developments" section before this 4.0 Applications section. We have also better described previous literature on evaluation and uncertainties associated with these data.

The intent of this section is to demonstrate the 'significance of the dataset' specifically the criteria that it is being used in a Climate Services/*decision support context for food and water security applications*, rather than a comprehensive evaluation.  This presentation is now better framed in the introduction where we now describe these data are motivated by the need for "Climate Services" where relative estimates and routinely updated information can be applied to different questions.

How do we know the system works for normal years or anomalous wet periods? How about for other regions outside the Afghanistan domain?

We do have some anecdotal examples of it working in wet periods (e.g. Widespread snowfall in Afghanistan). The reader could also refer to the products in Table 4. To confirm performance in normal or anomalous wet periods, the FEWS NET Afghanistan Seasonal Monitors highlight the use of these data since 2018.

However, we understand that the reviewer is likely hoping for a more quantitative analysis! We find that the development of a metric that would account for performance of a derived indicator is beyond the scope of this data descriptor e.g. categorical statistics (POD and FAR) for below-normal, normal, and above-normal years. This would require an independent reference dataset. We and other authors have attempted comparison with remotely sensed data (e.g. soil moisture, evaporation, total water storage, microwave snow estimates) but each of these data sources has its own set of errors that needs to be accounted for in the interpretation.

We hope that with the improvements to the introduction in terms of our motivation to provide inputs to Climate Services, as well as a review of prior evaluations better frames this section as a demonstration of the significance of these datasets, specifically that they are being applied in routine decision support.

12. Figure 10: Notice some of the text in the figure is too small to read.

We will separate out the figures so that the text is legible in the resubmission.

[Figure]

Figure 10a. Basins and provinces highlighted in the 2017-18 drought example.

[Figure]

Figure 10b. CHIRPS cumulative rainfall for 2017-18 vs average conditions for Daykundi Province. Figure from USGS EWX.

[Figure]

Figure 10c. CHIRPS cumulative rainfall for 2017-18 vs average conditions for Maydan Wardak Province. Figure from USGS EWX.

[Figure]

Figure 10d. Helmand Basin snow water equivalent (SWE) from the Central Asia data stream. The grey shading indicates the range of the minimum and maximum values, dashed blue line is the average, and black line is 2017-18. Figure from USGS EWX.

---

## Author Response (AR2)

June 7, 2022 – Author's response

Dear Editor,
Thank you for your comments and it's been a pleasure to work with ESSD. We have addressed the following points in the corrected submission.

1) We have added appropriate data access DOI (e.g. for CA as well as for global) at end of abstract.
2) We have corrected a few punctuation errors and will have a proofreader review at the proof stage.
3) In Section 4 the text and application are referring to the FLDAS Central Asia data stream's Snow Water Equivalent (SWE) anomalies in Figure 9 as well as the FLDAS Central Asia SWE timeseries in Figure 10c. Figure 10a & b show timeseries of the the CHIRPS precipitation which derived the FLDAS Global data stream. The MODIS SCF was only used in the evaluation section, and not used in Section 4.
4) In addition, we updated the multi-panel figures to be in the same file as required in the final upload section.

Sincerely,
Amy McNally & co-authors